# Learning Cascade Ranking as One Network

**Yunli Wang** [1]  **Zhen Zhang** [1]  **Zhiqiang Wang** [1]  **Zixuan Yang** [1]  **Yu Li** [1]  **Jian Yang** [2]  **Shiyang Wen** [1]  **Peng Jiang** [1]
**Kun Gai** [3]

## Abstract

Cascade Ranking is a prevalent architecture in large-scale top-k selection systems like recommendation and advertising platforms. Traditional training methods focus on single-stage optimization, neglecting interactions between stages. Recent advances have introduced interaction-aware training paradigms, but still struggle to 1) align training objectives with the goal of the entire cascade ranking (i.e., end-to-end recall of ground-truth items) and 2) learn effective collaboration patterns for different stages. To address these challenges, we propose LCRON, which introduces a novel surrogate loss function derived from the lower bound probability that ground truth items are selected by cascade ranking, ensuring alignment with the overall objective of the system. According to the properties of the derived bound, we further design an auxiliary loss for each stage to drive the reduction of this bound, leading to a more robust and effective top-k selection. LCRON enables end-to-end training of the entire cascade ranking system as a unified network. Experimental results demonstrate that LCRON achieves significant improvement over existing methods on public benchmarks and industrial applications, addressing key limitations in cascade ranking training and significantly enhancing system performance.

## 1. Introduction

Cascade ranking has emerged as a prevalent architecture in large-scale top-k selection systems, widely adopted in industrial applications such as recommendation and advertising platforms. This architecture efficiently balances resource

[1]Kuaishou Technology, Beijing, China [2]Beihang University, Beijing, China [3]Independent, Beijing, China. Correspondence to: Yunli Wang, Jian Yang <wangyunli@kuaishou.com, jiaya@buaa.edu.cn>.

*Proceedings of the 42nd International Conference on Machine Learning*, Vancouver, Canada. PMLR 267, 2025. Copyright 2025 by the author(s).

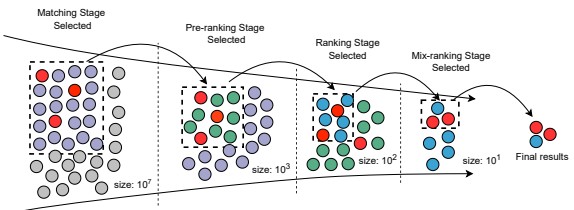

*Figure 1.* A typical cascade ranking architecture, including four stages: Matching, Pre-ranking, Ranking, and Mix-ranking. The red points represent the ground truth for the selection.

utilization and performance through a multi-stage, funnel-like filtering process. A typical cascade ranking system comprises multiple stages, including Matching, Pre-ranking, Ranking, and Mix-ranking, as illustrated in Figure 1. The objective is to select ground truth items (referred to as the red points in Figure 1) as the final outputs.

Early traditional training approaches for cascade ranking systems often optimize each stage in isolation, constructing samples, designing learning objectives, and defining proxy losses separately (Crammer & Singer, 2001; Burges et al., 2005; Li et al., 2007; Covington et al., 2016; Wang et al., 2018; Ma et al., 2018; Sheng et al., 2023; Wu et al., 2024). This fragmented approach overlooks the interactions between stages, leading to suboptimal alignment with the overall system objective. Specifically, two key challenges arise: 1) **Misalignment of Training Objectives**: Cascade ranking aims to collaboratively select all relevant items from the ground truth set across multiple stages. However, each stage's learning targets optimized by pointwise or pairwise losses are often more strict than the collaborative goal of cascade ranking. This misalignment may lead to inefficiency, particularly in recommendation scenarios where the model capacity is typically limited. 2) **Lack of Learning to Collaborate**: During online serving, different stages in a cascade ranking system will interact and collaborate with each other. Efficient interactions and collaborations are crucial for improving the overall performance of the cascade ranking system. For example, a Retrieval model can preemptively avoid items that the ranking model tends to overestimate, while the Ranking model can accurately identify ground-truth items from the recalled set. However, when models for different stages are trained independently,

they lack the ability to learn these interactions and collaborations, which may lead to degraded testing performance.

Recent works have attempted to address these challenges. ICC (Gallagher et al., 2019) is an early study to partially address challenges 1) and 2) by fusing the predicted scores of different stages and optimizing them jointly through LambdaRank (Burges, 2010). However, its stage-wise interaction remains unidirectional, restricting the learning of bidirectional collaboration. It also suffers from limited sample space, as it focuses only on exposed items. RankFlow (Qin et al., 2022) introduces an iterative training paradigm that dynamically determines the training samples for each stage by its upstream stage to address challenge 2). Although RankFlow reports significant improvements over both independent training and ICC, its iterative training process may lead to increased complexity and instability during training. FS-LTR (Zheng et al., 2024) tackles challenge 2) by learning online patterns of interactions and collaborations with full-stage training samples and outperforms RankFlow. However, neither RankFlow nor FS-LTR explicitly addresses the misalignment of training objectives. ARF (Wang et al., 2024) emphasizes that learning targets should align with the objective of cascade ranking and proposes a new surrogate loss for Recall based on differentiable sorting. However, ARF focuses on a single stage and cannot fully address challenges 1) and 2). Additionally, ARF does not fully utilize the information of the soft permutation matrix, further limiting its performance.

**To the best of our knowledge, no existing approach simultaneously addresses both challenges, highlighting the need for a more comprehensive solution.**

To address these challenges, we propose LCRON (**L**earning **C**ascade **R**anking as **O**ne **N**etwork), introducing two types of novel surrogate losses. First, we propose a novel surrogate loss $L_{e2e}$, which is the lower bound of a differentiable approximation of the survival probability of ground-truth items through the entire cascade ranking system. $L_{e2e}$ **directly aligns the learning objective with the global goal of the cascade ranking system, while naturally learning effective collaboration patterns for different stages**. However, optimizing $L_{e2e}$ alone may lead to insufficient supervision for individual stages, especially when the survival probability at a particular stage is close to 0. In addition, we derive the lower bound and find that the tightness of the bound is highly related to the consistency of different stages. Therefore, we design a surrogate loss $L_{single}$ towards the Recall of each single stage, which enforces the model to distinguish ground-truth items from the entire candidate set rather than the filtered subset from upstream stages. $L_{single}$ **can tighten the theoretical bound of $L_{e2e}$ and provides additional effective supervision when the survival probability of ground-truth items in $L_{e2e}$ is close to 0**. Inspired

by ARF (Wang et al., 2024), we use differentiable sorting techniques as the foundation of our surrogate losses. $L_{single}$ inherits the main idea of the $L_{Relax}$ loss in ARF while addressing its limited use of the soft permutation matrix generated by differentiable sorting. Finally, we combine different losses of LCRON in a UWL (Kendall et al., 2018) form to reduce the number of hyperparameters, enhancing its practicality and robustness.

To verify the effectiveness of our method, we conduct extensive experiments on both public and industrial benchmarks. We conduct public experiments on RecFlow (Liu et al., 2025), the only public benchmark based on a real-world recommendation system that contains multi-stage samples of cascade ranking. The results on the public benchmark show that LCRON outperforms all baselines under the streaming evaluation (where for any day $t$ as a test, its training data comes from the beginning up to day $t - 1$), indicating the effectiveness and robustness of our method. The ablation study highlights the role of different components of LCRON. The industrial experimental results show that LCRON consistently outperforms the best two baselines in public experiments on end-to-end Recall. We further conduct an online A/B experiment in a real-world advertising system. Compared to FS-LTR, **LCRON brings about a 4.10% increase in advertising revenue and a 1.60% increase in the number of user conversions, demonstrating that our approach has significant commercial value for real-world cascade ranking systems**.

## 2. Problem Formulation

We first introduce the formulation of a cascade ranking system. In such systems, a large initial set of candidate items is processed through a series of consecutive filtering stages to identify the most relevant or optimal results efficiently. Each stage applies a specific model $\mathcal{M}_i$ to evaluate and select a subset of items from the input set for the next stage. Let $T$ denote the total number of stages in the cascade, with an initial candidate inventory size of $q_0$. For any given stage $i$, we define the sample space of input candidates as $\mathcal{Q}_{i-1}$, which contains $q_{i-1}$ items. After processing by the $i$-th stage model, the output consists of $q_i$ selected items. Note that $q_i$ typically decreases as $i$ increases.

The models in the cascade ranking system are typically trained using specific paradigms, with training samples derived from the system itself. To rank the items, we define $\mathcal{F}_{\mathcal{M}}^{\downarrow}(\mathcal{S})$ as the ordered terms vector of set $\mathcal{S}$ sorted by the score of model $\mathcal{M}$ in descending order, and $\mathcal{F}_{\mathcal{M}}^{\downarrow}(\mathcal{S})[: K]$ as the top $K$ terms of $\mathcal{F}_{\mathcal{M}}^{\downarrow}(\mathcal{S})$. With the set of trained models $\{\mathcal{M}_i \mid 1 \leq i \leq T\}$, the final output set $CS_{out}$ of the cascade ranking system can be obtained through a sequential filtering operation. This process can be formulated as:

$$CS_{out} = \mathcal{F}^{\downarrow}_{\mathcal{M}_T}((\cdots \mathcal{F}^{\downarrow}_{\mathcal{M}_1}(\mathcal{Q}_0)[:q_1]\cdots))[:q_T] \quad (1)$$

We define the ground truth set $CS_{gt}$ as the set of items considered relevant or optimal based on user feedback or expert annotations. **The goal of training paradigms for cascade ranking is to optimize the Recall of $CS_{gt}$ using training data collected from the system**, which can be formulated as Eq 2, where $q_T$ is the size of $CS_{out}$, $\mathcal{K}$ is the size of $CS_{gt}$, and $\mathcal{K} < q_T$. Here, $\mathbf{1}(\cdot)$ is the indicator function that returns 1 if the condition is true and 0 otherwise.

$$Recall@\mathcal{K}@q_T = \frac{\sum\limits_{i=1}^{q_T} \mathbf{1}(item_i \in CS_{out})\mathbf{1}(item_i \in CS_{gt})}{\sum\limits_{j=1}^{\mathcal{K}} \mathbf{1}(item_j \in CS_{gt})} \quad (2)$$

Most previous works design training sets and methodologies for different stages separately, while a few attempt to build universal training paradigms, as detailed in Section 3. In this paper, we propose an all-in-one training paradigm for cascade ranking systems, which addresses the limitations of previous approaches and is detailed in Section 4.

## 3. Related Work

### 3.1. Learning Methodologies for Cascade Ranking

Cascade ranking (Wang et al., 2011; Li et al., 2023) is widely used in online recommendation and advertising systems to balance performance and resource efficiency. It employs multiple models with varying capacities to collaboratively select top-$k$ items from the entire inventory. Early traditional training approaches for cascade ranking systems often optimize each stage separately, with distinct training sample organization, learning objectives, and surrogate losses. There are three common learning tasks in cascade ranking systems: 1) **probability distribution estimation** (e.g., pCTR), which aims to optimize the accuracy of probability estimation and order and uses surrogate losses such as BCE, BPR, or their hybrid (Zhou et al., 2018; Ma et al., 2018; Sheng et al., 2023; Huang et al., 2022). The training samples include both positive and negative samples after exposure. 2) **continuous value estimation** (e.g., video playback time, advertising payment amount), which usually employs surrogate losses such as ordinal regression to optimize the model (Niu et al., 2016; Lin et al., 2023). It focuses on learning continuous values after specific user behaviors occur (e.g., viewing time after video exposure). 3) **learning-to-rank**, which is more commonly used in the retrieval stage of cascade ranking systems, with the entire or partial order of the ranking stage as the ground truth. It leverages methods such as pointwise, pairwise, and listwise approaches (Crammer & Singer, 2001; Li et al., 2007; Covington et al., 2016; Wang et al., 2018; Thonet et al., 2022; Tang et al., 2022; Wu et al., 2024; Wang

et al., 2024). While these methods are widely adopted, they often fail to align training objectives with the global goal of cascade ranking and overlook the interactions between different stages.

**Recently, several works have attempted to address these challenges by proposing interaction-aware training paradigms to jointly train the entire cascade ranking system**. ICC (Gallagher et al., 2019) fuses the predictions of different stages and optimizes the fusion score using LambdaRank (Burges, 2010). However, it suffers from limited sample space and unidirectional stage-wise interaction. RankFlow (Qin et al., 2022) introduces an iterative training paradigm. Each stage is trained with samples generated by its upstream stage and distills knowledge from its downstream model. While RankFlow reports significant improvements over ICC, its iterative training process may increase complexity and instability. FS-LTR (Zheng et al., 2024) argues that each stage model should be trained with full-stage samples. It trains the cascade ranking system using full-stage samples and LambdaRank loss, achieving better results than RankFlow. However, both FS-LTR and RankFlow fail to fully align with the global goal of cascade ranking. Another approach, ARF (Wang et al., 2024), emphasizes the importance of aligning learning targets with the Recall of cascade ranking and proposes surrogate losses based on differentiable sorting to optimize Recall. However, ARF focuses only on a single stage and assumes that downstream models are optimal, limiting its applicability. In this paper, we propose LCRON for end-to-end alignment with the global objective of cascade ranking, addressing the aforementioned challenges.

In addition, some recent works on cascade ranking systems offer complementary perspectives that could potentially be integrated with joint training approaches. FAA (Li et al., 2023) focuses on feature consistency of cascade ranking by aligning stage-wise representations via attention, while LCRON complements this approach by introducing end-to-end loss functions for global optimization. SRCR (Zamani et al., 2022) focuses on improving two-stage cascade systems but employs a non-learnable component (BM25) for retrieval and employs BERT (Devlin et al., 2019) for ranking and just jointly optimizes the number of retrieved documents $N$ and the ranking model. This differs from LCRON, which enables end-to-end joint learning across fully learnable cascade stages. These two approaches are conceptually complementary, suggesting that future work could explore joint optimization of both model parameters and system-level decision variables such as retrieval quota.

### 3.2. Differentiable Techniques for Hard Sorting

Differentiable sorting techniques provide continuous relaxations of the sorting operation, enabling end-to-end training

within deep learning frameworks. Early work by Grover et al.(2019) introduced NeuralSort, which approximates hard sorting by generating a unimodal row-stochastic matrix. Cuturi et al.(2019) further formulated the sorting process as an optimal transport problem with entropic regularization, enabling smooth approximations of ranks and sorted values. Subsequently, Prillo & Eisenschlos(2020) proposed Soft-Sort, a more lightweight and efficient approach for differentiable sorting. Recent advances, such as those in (Petersen et al., 2021; 2022; Sander et al., 2023; Cuturi et al., 2019), have further improved the performance and efficiency of differentiable sorting operators. These techniques have been widely adopted in various domains, including computer vision (Grover et al., 2019; Blondel et al., 2020; Sander et al., 2023), online recommendation and advertising (Swezey et al., 2021; Pobrotyn & Białobrzeski, 2021; Wang et al., 2024). In this paper, we leverage differentiable sorting operators to construct the surrogate losses of LCRON. It employs the soft permutation matrix produced by differentiable sorting methods. Thus, those differentiable sorting methods that do not produce a soft permutation matrix, such as Fast-Sort (Blondel et al., 2020) and OT (Cuturi et al., 2019), can not be the foundational component of LCRON.

## 4. Methodology

In this section, we introduce our proposed method LCRON, which is the abbreviation of "**L**earning **C**ascade **R**anking as **O**ne **N**etwork". In Section 4.1, we give the organization of full-stage training samples, which mainly follows Zheng et al.(2024). In Section 4.2, we introduce a novel loss $L_{e2e}$ that directly optimizes the lower bound of a differentiable approximation of Equation 2, ensuring alignment with the overall objective of cascade ranking. In section 4.3, we discuss the limitations of $L_{CS}$ and introduce $L_{single}$ as an auxiliary loss for each single stage to tighten the theoretical bound of $L_{e2e}$ and provide additional effective supervision.

### 4.1. Full-Stage Training Samples of Cascade Ranking

**For simplicity, we illustrate our method using a two-stage cascade ranking system ($T = 2$), which can be readily extended to systems with additional stages**. We downsample items from all stages of the cascade ranking system to construct full-stage training samples, primarily following FS-LTR (Zheng et al., 2024).

Let $\mathcal{M}_1$ and $\mathcal{M}_2$ denote the retrieval and ranking models, respectively. The input space of $\mathcal{M}_1$, the input and output spaces of $\mathcal{M}_2$ are $\mathcal{Q}_0$, $\mathcal{Q}_1$, and $\mathcal{Q}_2$. We denote the ground-truth set as $CS_{gt}$, also referred to as $\mathcal{Q}_3$. For a given impression, let $u$ represent the user information and $item_j$ represent the item information. Let $y_j$ denote the label for the pair $(u, item_j)$ and $\mathbf{y} = (y_1, y_2, \ldots, y_N)$. The training samples for one impression can be formulated as Eq 3:

$$D = (u, \{(item_j, y_j) \mid 0 \le j < N\})$$
$$= \left( u, \bigcup_{i=0}^{3} \{(item_j, y_j) \mid 0 \le j < n_i \text{ and } item_j \in \mathcal{Q}_i\} \right) \quad (3)$$

where $n_i$ is the number of samples drawn from $\mathcal{Q}_i$ and $N = n_0 + n_1 + n_2 + n_3$. In LCRON, the construction of full-stage training samples is a fundamental component of our approach. While specific sampling strategies and hyperparameters (e.g., $n_i$) may influence the performance, their detailed analysis is beyond the scope of this paper. In both public and online experiments, all methods are evaluated on the same sampled datasets to ensure a fair comparison.

Let $R_j$ denote the descending rank index (where a higher value indicates a better rank) for the pair $(u, item_j)$ within its sampled stage. The label $y_j$ is determined by both the stage order and the rank within the stage. Specifically, for any two pairs $(u, item_i)$ and $(u, item_j)$, the label $y_i$ and $y_j$ satisfy the following condition:

$$\mathbf{1}(y_i > y_j) = \mathbf{1}(\mathcal{S}_i > \mathcal{S}_j) \vee (\mathbf{1}(\mathcal{S}_i = \mathcal{S}_j) \wedge \mathbf{1}(R_i > R_j)) \quad (4)$$

where $\mathcal{S}_i > \mathcal{S}_j$ indicates that $item_i$ belongs to a later stage than $item_j$, and $R_i > R_j$ indicates that $item_i$ has a higher rank than $item_j$ within the same stage.

### 4.2. End-to-end Surrogate Loss for Top-K Cascade Ranking

Our goal is to design an efficient surrogate loss that aligns with the $Recall@\mathcal{K}@q_2$ metric of the entire cascade ranking system. We can transform the problem of optimizing the $Recall@\mathcal{K}@q_2$ of cascade ranking into a survival probability problem, allowing the model to estimate the probability that the ground truth is selected by the cascade ranking. Note that $\mathcal{K}$ is the size of $CS_{gt}$. Let $\mathcal{M}_i(D) \in \mathbb{R}^{1 \times N}$ denote the prediction vector of $\mathcal{M}_i$ on the training data $D$. Let $P_{\mathcal{M}_i}^{q_i}$ represent the probability vector of each ad in $D$ being selected by the cascade ranking for top-$q_i$ selection. $q_i$ is the quota of $\mathcal{M}_i$, as mentioned in Section 2. The sum of $P_{\mathcal{M}_i}^{q_i}$ should be $q_i$. Let the survival probability vector output by the system be $P_{CS}^{q_2}$, and let the label $\mathbf{y}$ be a binary vector indicating whether it is the ground truth. We can employ a cross-entropy loss of $P_{CS}^{q_2}$ and $\mathbf{y}$ to optimize the probability of the ground truth being selected by the cascade ranking as shown in Equation 5:

$$CE(P_{CS}^{q_2}, \mathbf{y}) = -\sum_i (y_i \ln((P_{CS}^{q_2})_i)$$
$$+ (1 - y_i) ln(1 - (P_{CS}^{q_2})_i) \quad (5)$$

Let $\pi \in \{0, 1\}^N$ denote the sampling result from $P_{\mathcal{M}_1}^{q_1}$, and

let $P_\pi$ represent the probability of sampling $\pi$. Then, $P_\pi$ can be formulated as Eq 6:

$$P_\pi = \frac{\prod_{i:\pi_i=1}(P^{q_1}_{\mathcal{M}_1})_i}{\sum_{S\subseteq[N],|S|=T}\prod_{j\in S}(P^{q_1}_{\mathcal{M}_1})_j} \tag{6}$$

Then $P^{q_2}_{CS}$ can be expressed as Eq 7:

$$P^{q_2}_{CS} = \mathbb{E}_{\pi\sim P_\pi}\frac{(P^{q_2}_{\mathcal{M}_2}\odot\pi)}{\langle\pi,P^{q_2}_{\mathcal{M}_2}\rangle/\langle\mathbf{1},P^{q_2}_{\mathcal{M}_2}\rangle} \tag{7}$$

where $P^{q_2}_{\mathcal{M}_2}$ and $P^{q_2}_{CS}$ are vectors. $\langle\cdot,\cdot\rangle$ denotes dot product, and $\odot$ denotes element-wise product. $\mathbf{1}$ represents a vector where all elements are 1. Due to the intractability of directly optimizing $P^{q_2}_{CS}$ caused by sampling and integration operations, we aim to find an approximate surrogate for $P^{q_2}_{CS}$. We define $\widehat{P^{q_2}_{CS}} = \prod_i^2 P^{q_i}_{\mathcal{M}_i}$. It can be shown that $\widehat{P^{q_2}_{CS}}$ serves as an lower bound for $P^{q_2}_{CS}$, as demonstrated in Eq. 8, since $0\le\langle\pi,P^{q_2}_{\mathcal{M}_2}\rangle/\langle\mathbf{1},P^{q_2}_{\mathcal{M}_2}\rangle\le 1$ always holds:

$$\begin{aligned}P^{q_2}_{CS} &= \mathbb{E}_{\pi\sim P_\pi}\frac{P^{q_2}_{\mathcal{M}_2}\odot\pi}{\langle\pi,P^{q_2}_{\mathcal{M}_2}\rangle/\langle\mathbf{1},P^{q_2}_{\mathcal{M}_2}\rangle}\\ &\ge \mathbb{E}_{\pi\sim P_\pi}P^{q_2}_{\mathcal{M}_2}\odot\pi\\ &= \prod_i^2 P^{q_i}_{\mathcal{M}_i}\\ &= \widehat{P^{q_2}_{CS}}\end{aligned} \tag{8}$$

The next question is how to obtain a differentiable $P^{q_i}_{\mathcal{M}_i}$, enabling us to optimize $\widehat{P^{q_2}_{CS}}$. To achieve this, we introduce the permutation matrix as the foundation of our approach. For a given vector $x$ and its sorted counterpart $y$, there exists a unique permutation matrix $\mathcal{P}$ such that $y = \mathcal{P}x$. The elements of $\mathcal{P}$ are binary, taking values of either 0 or 1. Let $\mathcal{P}^\downarrow_{(\cdot)}$ denote the permutation matrix that sorts the vector $(\cdot)$ in descending order. Specifically, $(\mathcal{P}^\downarrow_x)_{i,j}=1$ indicates that $x_j$ is the $i$-th largest element in $x$.

Using the permutation matrix, the top-$k$ elements of $D$ selected by $\mathcal{M}_i$ can be formulated as:

$$\mathcal{M}^\downarrow_i(k) = \sum_{j=1}^k(\mathcal{P}^\downarrow_{\mathcal{M}})_{j,:} \tag{9}$$

where $\mathcal{M}^\downarrow(k)\in\mathbb{R}^{1\times n}$ is a binary vector indicating whether each item is selected by the model, and $(\mathcal{P}^\downarrow_{\mathcal{M}})_{j,:}$ represents the $j$-th row of the permutation matrix for model $\mathcal{M}$.

It is evident that $P^{q_i}_{\mathcal{M}_i}$ can be interpreted as the distribution of $\mathcal{M}^\downarrow_i(q_i)$, where $\mathcal{M}^\downarrow_i(q_i)$ denotes the deterministic top-$q_i$

selection obtained through hard sorting (represented as a binary posterior observation, i.e., 1/0). To enable gradient-based optimization, we can relax $\mathcal{M}^\downarrow_i(q_i)$ into the stochastic $P^{q_i}_{\mathcal{M}_i}$ and optimize it via maximum likelihood. Specifically, this can be achieved by relaxing the hard permutation matrix $\mathcal{P}$ (which sorts items in descending order) into a soft permutation matrix $\hat{\mathcal{P}}$ via differentiable sorting techniques (Grover et al., 2019; Prillo & Eisenschlos, 2020; Petersen et al., 2021). Differentiable sorting methods typically generate $\hat{\mathcal{P}}$ as a row-stochastic matrix (each row sums to 1) by applying row-wise softmax with a temperature parameter $\tau$. As $\tau\to 0$, $\hat{\mathcal{P}}$ converges to the hard permutation matrix $\mathcal{P}$.

Specifically, let $\hat{\mathcal{P}}^\downarrow_{\mathcal{M}_i}\in[0,1]^{N\times N}$ be the soft permutation matrix for model $\mathcal{M}_i$, where $(\hat{\mathcal{P}}^\downarrow_{\mathcal{M}_i})_{j,k}$ represents the soft probability that item $k$ is ranked at position $j$. Then, we can formulate the top-$q_i$ selection probability $P^{q_i}_{\mathcal{M}_i}$ and the end-to-end loss $L_{e2e}$ of the cascade ranking system as Eq 10 and Eq 11:

$$P^{q_i}_{\mathcal{M}_i} = \frac{\sum_{j=1}^{q_i}(\hat{\mathcal{P}}^\downarrow_{\mathcal{M}_i})_{j,:}}{\oslash\, sp(\sum_{t=1}^{}(\hat{\mathcal{P}}^\downarrow_{\mathcal{M}_i})_{t,:})} \tag{10}$$

$$\begin{aligned}L_{e2e} = &-\sum_j y_j ln(\prod_i^2\frac{\sum_{j=1}^{q_i}(\hat{\mathcal{P}}^\downarrow_{\mathcal{M}_i})_{j,:}}{\oslash\, sp(\sum_{t=1}^{}(\hat{\mathcal{P}}^\downarrow_{\mathcal{M}_i})_{t,:})}\\ &-\sum_j(1-y_j)ln(1-\prod_i^2\frac{\sum_{j=1}^{q_i}(\hat{\mathcal{P}}^\downarrow_{\mathcal{M}_i})_{j,:}}{\oslash\, sp(\sum_{t=1}^{}(\hat{\mathcal{P}}^\downarrow_{\mathcal{M}_i})_{t,:})})\end{aligned} \tag{11}$$

where $\frac{\dot{\,}}{\oslash\dot{\,}}$ represents the element-wise division operator. In Eq 10, we perform an element-wise division to ensure that the probability values are normalized, as some differentiable sorting operators cannot guarantee column-wise normalization, such as NeuralSort (Grover et al., 2019) and SoftSort (Prillo & Eisenschlos, 2020). "$sp$" denotes that a stop gradient is needed during training. **The end-to-end loss $L_{e2e}$ in Eq. 11 directly optimizes the joint survival probability through all cascade stages**.

Traditional losses often impose strict constraints, which become problematic when model capacity is insufficient to satisfy all constraints. In contrast, $L_{e2e}$ **directly aligns the learning objective of the goal of cascade ranking, allowing models to prioritize critical rankings while tolerating minor errors in less important comparisons**. In addition, **when a stage assigns a low score to a ground-truth item, $L_{e2e}$ not only optimizes that particular stage but also encourages other stages to improve their scores for the same ground-truth item. This mechanism enables an efficient collaborative pattern to be learned across stages**, enhancing the overall survival probability of ground-truth items in the cascade ranking system. Importantly, this collaboration is bidirectional, overcoming the limitation of

ICC (Gallagher et al., 2019), which only allows unidirectional dependency from one stage to its pre-stages (e.g., from the ranking stage to the retrieval stage).

## 4.3. Auxiliary Loss of Single Stage for Tightening the Bound

Although $L_{e2e}$ directly optimizes the joint survival probability of ground-truth items through the entire cascade ranking system, it may suffer from two weaknesses: 1) $L_{e2e}$ is derived as a lower bound of the joint survival probability (see Section 4.2), the tightness of this bound could influence the overall performance, but $L_{e2e}$ can not optimize this bound itself, 2) it may suffer from insufficient supervision when the survival probability at a particular stage is close to 0. This issue is particularly pronounced during the initial training phase, where if all stages assign low scores to ground-truth items, the gradients across all models would be small. This situation could make it difficult to properly warm up the models, resulting in slow convergence or suboptimal learning performance.

According to Eq 8, the bound depends on the magnitude of $\frac{q_2}{\langle \pi, P_{\mathcal{M}_2}^{q2} \rangle}$, and the equation holds as $\frac{q_2}{\langle \pi, P_{\mathcal{M}_2}^{q2} \rangle}$ approaches 1. Since $\mathbb{E}_{\pi \sim P_\pi}(\pi) = P_{\mathcal{M}_1}^{q1}$, it is evident that optimizing the difference between $\mathcal{M}_1(D)$ and $\mathcal{M}_2(D)$ can help tighten the bound. We provide a detailed analysis in appendix A.

To address these issues, we propose $L_{single}^{\mathcal{M}_i}$ for each single model, which is shown in Eq 12. $L_{single}$ **provides the same supervision for both retrieval and ranking models, thereby optimizing the consistency of** $\mathcal{M}_1(D)$ **and** $\mathcal{M}_2(D)$ **and leading to a tight bound of** $L_{e2e}$. $L_{single}^{\mathcal{M}_i}$ forces each model to distinguish ground-truth items from the entire inventory, offering effective yet not overly strict additional supervision for each model. This helps address the insufficient supervision situation that may occur in the $L_{e2e}$ loss during the initial training phase. Furthermore, unlike $L_{Relax}$ in ARF, which uses only the top-$m$ rows of the soft permutation matrix, $L_{single}$ imposes supervision signals over the entire soft permutation matrix. This provides more comprehensive supervision information and is expected to facilitate better model learning.

$$
\begin{aligned}
L_{single}^{\mathcal{M}_i} = &- \sum_j y_j ln\Big(\frac{\sum_{j=1}^{\mathcal{K}}(\hat{\mathcal{P}}_{\mathcal{M}_i}^{\downarrow})_{j,:}}{\oslash sp(\sum_{t=1}^{\mathcal{K}}(\hat{\mathcal{P}}_{\mathcal{M}_i}^{\downarrow})_{t,:})}\Big) \\
&- \sum_j (1 - y_j) ln\Big(1 - \frac{\sum_{j=1}^{\mathcal{K}}(\hat{\mathcal{P}}_{\mathcal{M}_i}^{\downarrow})_{j,:}}{\oslash sp(\sum_{t=1}^{\mathcal{K}}(\hat{\mathcal{P}}_{\mathcal{M}_i}^{\downarrow})_{t,:})}\Big)
\end{aligned}
\tag{12}
$$

Inspired by ARF (Wang et al., 2024), we simply employ UWL (Kendall et al., 2018) to balance the $L_{e2e}$ and $L_{single}$ to reduce the number of hyper-parameters. For the two-stage cascade ranking, the final loss is formulated as Eq 13, where $\alpha$, $\beta$ and $\gamma$ are trainable scalars.

*Table 1.* Dataset Statistics.

| Stage | Users | Impressions | Items per impression | the range of labels |
|---|---|---|---|---|
| rank_pos | 38,193 | 6,062,348 | 10 | [1,20] |
| rank_neg | 38,193 | 6,062,348 | 10 | [21,21] |
| coarse_neg | 38,193 | 6,062,348 | 10 | [22,22] |
| prerank_neg | 38,193 | 6,062,348 | 10 | [23,23] |

$$
L = \frac{L_{e2e}}{2\alpha^2} + \frac{L_{single}^{\mathcal{M}_1}}{2\beta^2} + \frac{L_{single}^{\mathcal{M}_2}}{2\gamma^2} + log_2(\alpha\beta\gamma)
\tag{13}
$$

## 5. Experiments

### 5.1. Experiment Setup

We conduct comprehensive experiments on both public and industrial datasets. We conduct public experiments to verify the effectiveness of our proposed method and perform ablation studies along with in-depth analysis. We conduct online experiments to study the impact of our method on real-world cascade ranking applications. Here we mainly describe the setup for public experiments, and details of online experiments are described in section 5.5.

- **Public Benchmark.** We conduct public experiments based on RecFlow (Liu et al., 2025), which, to the best of our knowledge, is the only public benchmark that collects data from all stages of real-world cascade ranking systems. RecFlow includes data from two periods (denoted as Period 1 and Period 2), spanning 22 days and 14 days, respectively. While Period 2 is primarily designed for studying the distribution shift problem in recommendation systems, which is beyond the scope of this paper, we focus on Period 1 data as our testbed. To train the two-stage cascade ranking, we adopt four stages of samples: $rank\_pos$, $rank\_neg$, $coarse\_neg$, and $prerank\_neg$. Table 1 summarizes the dataset statistics.

- **Cascade Ranking Setup.** We employ a typical two-stage cascade ranking system as the testbed, utilizing DIN (Zhou et al., 2018) for the ranking model and DSSM (Huang et al., 2013) for the retrieval model. The retrieval and ranking models are completely parameter-isolated, ensuring no parameters are shared between stages. This design eliminates potential confounding effects from parameter sharing, enabling a fair comparison between different methods. In this setup, the retrieval model selects the top 30 items, and subsequently, the ranking model chooses the top 20 items out of these 30 for "exposure". Each impression contains 10 ground-truth items, referred to $rank\_pos$, which serve as the ground truth for exposure evaluation. This setup aligns with the data structure of the benchmark, ensuring evaluation consistency with real-world cascade ranking scenarios.

- **Evaluation. We employ** $Recall@k@m$ **defined in Equation 2 as the golden metric** to evaluate the overall per-

formance of the cascade ranking system. Corresponding to the cascade ranking and public benchmark setup, the $m$ and $k$ for the evaluation are 20 and 10, respectively. In order to explore the impact of different baselines on model learning at different stages, we also evaluate the $Recall@k@m$ and $NDCG@k$ of each model on the entire inventory of candidate items as auxiliary observation metrics for analysis.

Following the mainstream setup for evaluating recommendation datasets, we use the last day of Period 1 as the test set to report the main results of our experiments (Section 5.3), while the second-to-last day serves as the validation set for tuning the hyperparameters (e.g., the temperature parameters of ICC, ARF, and LCRON; the $\alpha$ of RankFlow; the top-k and smooth factor of FS-LambdaLoss) of different methods. Considering that industrial scenarios commonly employ streaming training, we further evaluate the performance of different methods by treating each day as a separate test set (in Section 5.4). In this setting, when day $t$ is designated as the test set, the corresponding training data encompass all days from the beginning of Period 1 up to day $t - 1$.

- **LCRON Setup.** We employ NeuralSort (Grover et al., 2019) as the differentiable sorting operator, aligning with the baseline method ARF to ensure a fair comparison. We tune the hyper-parameter $\tau$ on the validation set, which controls the temperature of NeuralSort. We set $q_1$ and $q_2$ in $L_{e2e}$ to 10 during training. Although this value does not exactly match the quotas of the cascade ranking setup, it represents a trade-off between gradient optimization stability and maintaining the interpretability of the loss due to the properties of the permutation matrix. A larger $q_i$ is more likely to lead to gradient conflicts during training. We further provide a detailed discussion of this limitation in Section E.1.

- **Implementation Details.** We utilize the training pipeline and implementations of DSSM (Huang et al., 2013) and DIN (Zhou et al., 2018) models provided by the open-source code[1] from (Liu et al., 2025), and we primarily focus on implementing the training loss functions of baselines and our method. The Multi-layer Perceptron (Rosenblatt, 1958) of the user and item towers in DSSM are set to be [128, 64, 32]. The architecture of DIN's MLP is [128, 128, 32, 1]. All offline experiments are implemented using PyTorch 1.13 in Python 3.7. We employ the Adam optimizer with a learning rate of 0.01 for training all methods. Following the common practice in online recommendation systems (Liu et al., 2025; Zhang et al., 2022), each method is trained for only one epoch. The batch size is set to 1024. The source code of our public experiments is publicly available[2].

---

[1]https://github.com/RecFlow-ICLR/RecFlow
[2]https://github.com/Kwai/LCRON

## 5.2. Competing Methods

We compare our method with the following state-of-the-art methods in previous studies.

- **Binary Cross-Entropy (BCE).** We treat the "rank_pos" samples as positive and others as negative to train a BCE loss. The retrieval model is trained with all stages samples in Table 1. The ranking model is trained with only rank_pos and rank_neg samples, following the classic setting of cascade ranking systems. The rank_pos is regarded as positive samples, and others are regarded as negative samples. It is denoted as "BCE" in the following.

- **ICC.** It's an early study for joint learning the models of cascade ranking (Gallagher et al., 2019), which fuse multi-stage predictions and optimize them by LambdaRank (Burges, 2010).

- **RankFlow.** Qin et al.(2022) propose RankFlow, a method that iteratively updates the retrieval and ranking models, achieving better results than ICC (Gallagher et al., 2019). The training sample space for the ranking model is determined by the retrieval model, and the ranking model's predictions are distilled back to the retrieval model.

- **FS-LTR.** Zheng et al.(2024) propose a method to train all models in a cascade ranking system using full-stage training samples and learning-to-rank surrogate losses, achieving better results than RankFlow (Qin et al., 2022). Since Zheng et al.(2024) primarily focuses on the organization of training samples rather than the design of surrogate losses, we design two representative variants to cover a wide spectrum of LTR losses. The first variant, **FS-RankNet**, utilizes the RankNet (Burges et al., 2005) loss, a classic pairwise learning-to-rank method. The second variant, **FS-LambdaLoss**, utilizes the LambdaLoss (Wang et al., 2018) loss, an advanced listwise LTR method. These variants represent two fundamental paradigms in LTR, ensuring a comprehensive comparison.

- **ARF.** It is designed to adapt to varying model capacities and data complexities by introducing an adaptive target that combines "Relaxed" loss and "Global" loss (Wang et al., 2024). We use ARF loss to train both the retrieval and ranking models. To further improve ARF, we introduce **ARF-v2** as an enhanced baseline, which replaces the relaxed loss of ARF with our proposed $L_{single}$ (Section 4.3). We set the hyper-parameters corresponding to the $q$ of the cascade ranking system. For the retrieval model, we set the parameters $m$ and $k$ to 30 and 10, respectively. For the ranking model, we set these parameters to 20 for $m$ and 10 for $k$.

## 5.3. Main Results

Table 2 presents the main results of the public experiments conducted on RecFlow. LCRON significantly outperforms

*Table 2.* Main results of public experiments on RecFlow. Each method was run 5 times, and the results are reported as mean±std. ∗ indicates the best results. Bold numbers indicate that LCRON shows statistically significant improvements over the baselines, as determined by a t-test at the 5% significance level. Note that the $Recall@10@20$ of $Joint$ is the golden metric for the whole cascade ranking system. The test set is the last day, with the remaining data used for training.

| Method/Metric | Joint | Ranking | | Retrieval | |
|---|---|---|---|---|---|
| | Recall@10@20 ↑ | Recall@10@20 ↑ | NDCG@10 ↑ | Recall@10@30 ↑ | NDCG@10 ↑ |
| BCE | 0.8539±0.0006 | 0.8410±0.0007 | 0.7043±0.0008 | 0.9706±0.0004* | 0.7150±0.0019 |
| ICC | 0.8132±0.0003 | 0.8100±0.0003 | 0.6980±0.0003 | 0.9288±0.0003 | 0.6155±0.0003 |
| RankFlow | 0.8647±0.0007 | 0.8629±0.0006 | 0.7274±0.0010 | 0.9656±0.0006 | 0.7087±0.0003 |
| FS-RankNet | 0.7881±0.0007 | 0.7908±0.0008 | 0.6864±0.0004 | 0.9321±0.0004 | 0.6710±0.0005 |
| FS-LambdaLoss | 0.8666±0.0016 | 0.8660±0.0018 | 0.7306±0.0027 | 0.9691±0.0004 | 0.7190±0.0027* |
| ARF | 0.8608±0.0006 | 0.8616±0.0007 | 0.6655±0.0027 | 0.9631±0.0008 | 0.5437±0.0110 |
| ARF-v2 | 0.8678±0.0009 | 0.8679±0.0009 | 0.7269±0.0005 | 0.9684±0.0006 | 0.7152±0.0028 |
| LCRON (ours) | **0.8732±0.0005*** | **0.8729±0.0004*** | 0.7291±0.0008 | 0.9700±0.0004 | 0.7151±0.0009 |

all baselines on the end-to-end (joint) Recall, demonstrating its effectiveness in optimizing cascade ranking systems. Notably, while LCRON does not dominate all individual stage metrics (e.g., the Recall of the Ranking model), it achieves substantial improvements in joint metrics, highlighting the importance of stage collaboration. This aligns with our design philosophy: LCRON fully considers the interaction and collaboration between different stages, enabling significant gains in joint performance even with modest improvements in single-stage metrics.

The results also reveal interesting insights about the baselines: 1) While FS-LambdaLoss shows strong performance, FS-RankNet performs poorly under the same sample organization. This indicates that the choice of learning-to-rank methods plays a critical role in optimizing cascade ranking systems. 2) ARF-v2 outperforms ARF in all metrics, suggesting that $L_{single}$ effectively mitigates the disadvantages of the $L_{Relax}$ loss in ARF, as discussed in Section 4.3.

### 5.4. In-depth Analysis

We first conduct an ablation study to evaluate the contribution of each component in LCRON. Specifically, we analyze the impact of $L_{e2e}$ and $L_{single}$ by removing them individually. The results are summarized in Table 3. The ablation study demonstrates that both $L_{e2e}$ and $L_{single}$ are essential components of LCRON. While their removal may not lead to statistically significant changes in all the individual ranking or retrieval metrics, the joint evaluation metric consistently shows a statistically significant improvement. This suggests that the interaction effects between the two stages of the cascade ranking system are optimized more effectively with the collaboration of $L_{e2e}$ and $L_{single}$, enhancing the overall performance.

In Section 5.3, we report the results tested on the last day of the dataset. Although this is a mainstream test setting for recommendation benchmarks (Zheng et al., 2024; Wang et al., 2024; Liu et al., 2025), we argue that it may not fully

*Table 3.* Ablation study of LCRON. ∗ indicates the best results. Each method was run 5 times, and the results are reported as $mean \pm std$. Bold numbers indicate that LCRON shows statistically significant improvements over each ablation model, as determined by a t-test at the 5% significance level. The test set is the last day, with the remaining data used for training.

| Method/Metric | Joint | Ranking | | Retrieval | |
|---|---|---|---|---|---|
| | Recall@10@20 ↑ | Recall@10@20 ↑ | NDCG@10 ↑ | Recall@10@30 ↑ | NDCG@10 ↑ |
| LCRON | **0.8732** ± 0.0005 * | **0.8729** ± 0.0004* | 0.7291 ± 0.0008 * | 0.9700 ± 0.0003 * | 0.7151 ± 0.0009 |
| $-L_{e2e}$ | 0.8710 ± 0.0013 | 0.8707 ± 0.0012 | 0.7280 ± 0.0007 | 0.9695 ± 0.0007 | 0.7153 ± 0.0009* |
| $-L_{single}$ | 0.8712 ± 0.0004 | 0.8712 ± 0.0005 | 0.7286 ± 0.0007 | 0.9692 ± 0.0006 | 0.7142 ± 0.0013 |

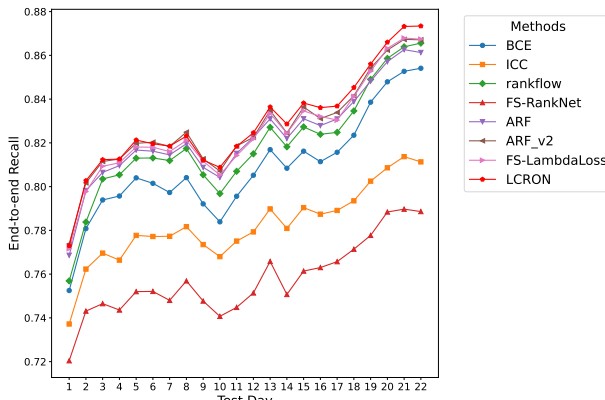

*Figure 2.* The evaluation results of different methods on RecFlow, in a streaming manner.

reflect real-world industrial scenarios, where models are typically trained in an online, streaming manner. To provide a more comprehensive evaluation, we conduct additional experiments under a streaming training setup, where each day is treated as a separate test set. Specifically, when day $t$ is designated as the test set, the training data encompass all days from the beginning up to day $t − 1$. The results, shown in Figure 2, reveal two key observations: 1) In the initial phase (first 10 days), LCRON exhibits rapid convergence, matching the performance of ARF-v2 and significantly outperforming other baselines. 2) In the later phase (last 12 days), LCRON not only surpasses all baselines but also demonstrates a widening performance gap over time, indicating its superior adaptability and robustness in long-term industrial applications. These findings underscore LCRON's ability to achieve both fast convergence and sustained performance improvements, making it a practical and effective solution for real-world cascade ranking systems.

Furthermore, we investigate the impact of manually tuning the weights of $L_{e2e}$ and $L_{single}$ in LCRON. While the UWL formulation is designed to reduce hyperparameter tuning costs, it does not guarantee optimal performance theoretically. Table 4 shows the results of fixing $L_{e2e}$ to 1 and varying the weight of $L_{single}$. We observe that different weight configurations yield varying performance, with most configurations outperforming the baselines. The best manual configuration achieves a joint $Recall@10@20$ of 0.8733,

*Table 4.* Experimental results of LCRON under fixed-weight configurations of $L_{e2e}$ and $L_{single}$. We fix $L_{e2e}$ to 1 and evaluate different weights for $L_{single}$. The test set is the last day, with the remaining data used for training.

| weight of $L_{single}$ | Joint | Ranking | | Retrieval | |
|---|---|---|---|---|---|
| | Recall@10@20 ↑ | Recall@10@20 ↑ | NDCG@10 ↑ | Recall@10@30 ↑ | NDCG@10 ↑ |
| 0.01 | 0.8712 | 0.8713 | 0.7294 | 0.9691 | 0.7122 |
| 0.1 | 0.8706 | 0.8706 | 0.7292 | 0.9686 | 0.7109 |
| 0.5 | 0.8705 | 0.8703 | 0.7279 | 0.9694 | 0.7132 |
| 1 | 0.8723 | 0.8723 | 0.7284 | 0.9691 | 0.7136 |
| 2 | 0.8731 | 0.8731 | 0.7295 | 0.9690 | 0.7132 |
| 3 | 0.8730 | 0.8730 | 0.7289 | 0.9694 | 0.7120 |
| 4 | 0.8720 | 0.8718 | 0.7293 | 0.9689 | 0.7118 |
| 5 | 0.8733 | 0.8730 | 0.7299 | 0.9697 | 0.7140 |

*Table 5.* Industrial experimental results for 15 days on a real-world advertising system. Each method was allocated 10% of the online traffic. For online metrics, we calculate the relative improvement of other methods compared to FS-LambdaLoss as the baseline.

| Method/Metric | Offline Metrics | Online Metrics | |
|---|---|---|---|
| | Joint Recall | Revenue | Ad Conversions |
| FS-LambdaLoss | 0.8210 | – | – |
| ARF-v2 | 0.8237 | +1.66% | +0.65% |
| LCRON (ours) | 0.8289 | +4.1% | +1.6% |

which is almost the same as UWL-based LCRON. This suggests that while manual tuning can yield competitive results, UWL provides a robust and efficient way to combine $L_{e2e}$ and $L_{single}$ without extensive hyperparameter search.

Overall, these in-depth analyses, including the ablation study, streaming evaluation, and weight tuning experiments, collectively demonstrate the robustness and practicality of LCRON in real-world cascade ranking systems. The ablation study confirms the necessity of both $L_{e2e}$ and $L_{single}$ for optimizing the interaction effects between ranking and retrieval stages. The streaming evaluation results further validate LCRON's ability to adapt to dynamic, real-world scenarios, while the weight tuning experiments highlight the efficiency of the UWL (Kendall et al., 2018) formulation in reducing hyperparameter tuning efforts without sacrificing performance. These findings solidify LCRON as a strong candidate for industrial applications, offering a balanced combination of performance, adaptability, and ease of deployment.

In addition, we conducted several supplementary experiments to further validate the effectiveness and robustness of LCRON under various settings. These include studies on cascade ranking systems with more than two stages ($T > 2$), the impact of different differentiable sorting operators, sensitivity analysis on hyperparameter $\tau$, as well as performance under varying learning rates and batch sizes. These experiments not only demonstrate LCRON's scalability to multi-stage cascade ranking systems (see Appendix C.2), but also its compatibility with alternative differentiable sorting techniques (see Appendix C.3). Moreover, LCRON exhibits stable performance across a wide range of $\tau$ values (see Appendix C.4) and consistently outperforms baselines under different training configurations (see Appendix C.5), highlighting its practicality and robustness for real-world applications. Due to space constraints, detailed settings, results, and analysis are provided in Appendix C.

### 5.5. Online Deployment

To study the impact of LCRON on real-world industrial applications, we further deploy LCRON in the advertising system of Kuaishou Technology. Due to the scarcity of online

resources, we opted to select only the two best-performing baselines from the public experiment, along with LCRON for the online A/B test. Each experimental group was allocated 10% of the online traffic. Each model was trained using an online learning approach and was deployed online after seven days of training, followed by a 15-day online A/B test. Due to the space limitation, implementation details are described in appendix B. The results of the online experiments, as shown in Table 5, demonstrate that our LCRON model achieves significant improvements in both revenue and ad conversions compared to the two baseline methods, FS-LambdaLoss and ARF-v2. Notably, LCRON delivers superior performance in both public and industrial experiments, where variations in data size and model architecture are present. This indicates not only the effectiveness but also the robust generalization capabilities of our approach. Based on the results of rigorous A/B testing, LCRON has been fully rolled out on the Kuaishou advertising platform since January 2025. The two models trained under LCRON have successfully replaced the primary pathways (i.e., those with the highest weight) in the Matching and Pre-ranking stages, marking a major milestone in its industrial deployment.

## 6. Conclusion

In this paper, we present LCRON, a novel training framework for cascade ranking systems, which incorporates two complementary loss functions: $L_{e2e}$ and $L_{single}$. $L_{e2e}$ optimizes a lower bound of the survival probability of ground-truth items throughout the cascade ranking process, ensuring theoretical consistency with the system's global objective. To address the limitations of $L_{e2e}$, we introduce $L_{single}$, which tightens the theoretical bound and provides additional supervisory signals to enhance stage-wise learning. We conduct extensive experiments on both public (RecFlow) and industrial benchmarks, as well as online A/B testing, demonstrating that LCRON significantly improves end-to-end Recall, advertising revenue, and user conversion rates. This work has the potential to make a broad positive impact across a range of applications, such as recommendation, advertising, and search systems. We also provide an in-depth discussion of the limitations and potential future directions of this work in Appendix E.

## Acknowledgements

We thank Lili Mou for his insightful discussion and suggestions on the early study of this work. We thank Jiangwei Guo, Daqing Chang, Qi Chen, Yangrui Wang, Hang Chen, Xiang He, Miaochen Li, Jian Hou, Qinglong Li, and Qian He for their contributions to the construction of data streams, enabling us to build full-stage training samples. We also thank Kai Zheng and Qi Liu for their assistance with the RecFlow benchmark, and Jupan Li, Jianghua You, Caiyi Xu, and Xu He for their help with the training and serving pipeline. Finally, we sincerely appreciate the valuable feedback and comments from the anonymous reviewers, which have helped improve the clarity and quality of the paper.

## Impact Statement

This paper presents work whose goal is to advance the field of Machine Learning. There are many potential societal consequences of our work, none which we feel must be specifically highlighted here.

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

# A. Theoretical Analysis for the Gap Between $P_{CS}^{q_2}$ and $\widehat{P_{CS}^{q_2}}$

Here, we provide a more detailed theoretical analysis of the gap between $P_{CS}^{q_2}$ and its lower bound $\widehat{P_{CS}^{q_2}}$, and explain how $L_{single}$ contributes to tightening this bound.

The gap between $P_{CS}^{q_2}$ and $\widehat{P_{CS}^{q_2}}$ can be formulated as Eq. 14:

$$
\begin{aligned}
\Delta &= P_{CS}^{q_2} - \widehat{P_{CS}^{q_2}} \\
&= \mathbb{E}_{\pi \sim P_\pi} \left[ \frac{P_{\mathcal{M}_2}^{q_2} \odot \pi}{\langle \pi, P_{\mathcal{M}_2}^{q_2} \rangle / q_2} - P_{\mathcal{M}_2}^{q_2} \odot \pi \right] \\
&= \mathbb{E}_{\pi \sim P_\pi} \left[ P_{\mathcal{M}_2}^{q_2} \odot \pi \left( \frac{q_2}{\langle \pi, P_{\mathcal{M}_2}^{q_2} \rangle} - 1 \right) \right] \\
&\leq \mathbb{E}_{\pi \sim P_\pi} \left[ P_{\mathcal{M}_2}^{q_2} \left( \frac{q_2}{\langle \pi, P_{\mathcal{M}_2}^{q_2} \rangle} - 1 \right) \right] \\
&= \left[ P_{\mathcal{M}_2}^{q_2} \left( \frac{q_2}{\mathbb{E}_{\pi \sim P_\pi} \langle \pi, P_{\mathcal{M}_2}^{q_2} \rangle} - 1 \right) \right] \\
&= \left[ P_{\mathcal{M}_2}^{q_2} \left( \frac{q_2}{\langle P_{\mathcal{M}_1}^{q_1}, P_{\mathcal{M}_2}^{q_2} \rangle} - 1 \right) \right] \\
&= \Delta'
\end{aligned}
\tag{14}
$$

Here, we have derived a theoretical upper bound for $\Delta$, denoted as $\Delta'$. Note that $P_{\mathcal{M}_2}^{q_2} \in [0,1]^N$, $P_{\mathcal{M}_1}^{q_1} \in [0,1]^N$, and $\sum P_{\mathcal{M}_2}^{q_2} = q_2$, $\sum P_{\mathcal{M}_1}^{q_1} = q_1$.

Next, we consider how the relationship between the model outputs might affect $\Delta'$. For a given $P_{\mathcal{M}_2}^{q_2}$, treating $P_{\mathcal{M}_2}^{q_2}$ as the only variable to minimize $\Delta'$, it is evident that the following condition must be satisfied:

$$
(P_{\mathcal{M}_1}^{q_1})_i = \begin{cases} 1 & \text{if } i \in \underset{i}{\arg\text{TopK}}(P_{\mathcal{M}_2}^{q_2}) \quad (\text{K=}q_1) \\ 0 & \text{otherwise} \end{cases}
\tag{15}
$$

where $(P_{\mathcal{M}_1}^{q_1})_i$ represents the $i$-th element of the vector $P_{\mathcal{M}_1}^{q_1}$, and $\underset{i}{\arg\text{TopK}}(P_{\mathcal{M}_2}^{q_2})$ denotes the indices of the top $K$ elements of $P_{\mathcal{M}_2}^{q_2}$. This indicates that if the top $q_2$ sets of the two models are consistent, it helps to reduce $\Delta'$. Of course, optimizing the consistency of the entire output distributions across different models also achieves the same goal, as this is a stronger constraint. Furthermore, if $(P_{\mathcal{M}_1}^{q_1})_i$ is a binary vector (i.e., its elements are either 0 or 1), then $\Delta$ achieves its minimum value ($\Delta' = 0$) when Eq. 15 is satisfied.

The single-stage loss $L_{single}^{\mathcal{M}_i}$ (Eq. 12) directly optimizes the ranking consistency of each model with the same supervision (ground-truth labels), thereby implicitly aligning $\mathcal{M}_1(D)$ and $\mathcal{M}_2(D)$. This helps to reduce $\Delta'$, indirectly optimizing the bound $\Delta$. It is worth noting that there are many methods to optimize the consistency of model outputs, such as model distillation or minimizing the KL divergence between outputs. The $L_{single}$ we designed not only optimizes the consistency of output distributions across different models but also mitigates potential gradient vanishing issues in $L_{e2e}$ under certain circumstances, providing additional effective supervision signals.

# B. Implementation Details of Online Experiments

In this section, we provide additional details on the implementation of the online experiments. Our online system consists of four stages: Matching, Pre-ranking, Ranking, and Mix-ranking. However, for the purpose of this study, we focus on a two-stage cascade ranking setup, comprising the Matching (Retrieval) and Pre-ranking stages (illustrated in Figure 1). This design choice is motivated by several practical and methodological considerations: 1) Generality of Two-Stage Setup: The two-stage cascade ranking setup (Matching and Pre-ranking) does not lose generality, as it captures the core challenges of cascade ranking optimization, which are fundamental to any multi-stage ranking system. The insights gained from this setup

can be extended to systems with more stages. 2) Practical Constraints: Conducting experiments on the entire four-stage system would introduce significant engineering challenges, particularly in data construction and model deployment. For instance, aligning the format of training logs and deployment details across all stages is non-trivial. Deploying and evaluating a full four-stage cascade ranking system in an online environment would require substantial infrastructure support, which is beyond the scope of this study.

Regarding the features, we utilize approximately 150 sparse features and 4 dense features. Sparse features are represented by embeddings derived from lookup tables, with each ID embedding having a dimensionality of 64. Dense features, on the other hand, directly use raw values or pre-trained model outputs, with a total dimensionality of 512. The Retrieval model includes all features used in the Pre-ranking model, except for combined features. Combined features refer to the process of integrating multiple individual features into a new, unified feature set to improve model performance. This technique captures interactions between different variables, providing richer information than each feature could individually. The combined features also fall under the category of sparse features. In our experiments, there are 20 combined features. In our experiments, models of different stages do not share any parameters.

Regarding the model architectures, we adopt DSSM (Huang et al., 2013) for the retrieval stage and MLP (Rosenblatt, 1958) for the Pre-ranking stage. For DSSM (Huang et al., 2013), the FFN layers' size of both the user and item towers is [1024,256,256,64]. The layer size of the MLP (Rosenblatt, 1958) is [1024,512,512,1]. We employ PRelu (He et al., 2015) as the activation function for hidden layers. We use HeInit (He et al., 2015) to initialize all the training parameters. We adopt batch normalization for each hidden layer, and the normalization momentum is 0.999.

Regarding the training samples, we organize our training samples according to Section 4.1. We collect samples from the Matching, Pre-ranking, and Ranking stages, with items that succeed in the Ranking stage treated as ground-truth items. For each impression, we collect 20 items for training (i.e., $N = 20$ in Section 4.1). The sample distribution across stages is defined by $n_0 = 5$, $n_1 = 5$, $n_2 = 8$, and $n_3 = 2$.

The models are trained in an online streaming manner. During each day of training, 20 billion (user, ad) pairs were processed. The optimizer is AdaGrad with a learning rate of 1e-2. All parameters are trained from scratch, without any pre-trained embeddings. The batch size is set to 4096 for both the retrieval and Pre-ranking models. Both the training and serving frameworks are developed based on TensorFlow. We use LCRON to train the retrieval and pre-rank model together and save unified checkpoints. Subsequently, by reconstructing two metadata files, these two models are deployed separately. During deployment, each model loads only the parameters corresponding to its own structure from the saved checkpoint during the joint training phase.

## C. More Experimental Details and Results on the Public Benchmark

### C.1. Implementation and Hyper-parameters Tuning Details of Baselines

Since none of the baseline methods have been evaluated on the RecFlow (Liu et al., 2025) dataset under cascade ranking settings, all baseline results were obtained by re-implementing or adapting the source code under the same experimental conditions as our proposed LCRON, rather than directly citing results from previous papers. For FS-RankNet and FS-LambdaLoss, we adapt standard implementations from the TF-Ranking library to PyTorch versions. For other baselines, when open-source code was available and runnable, we used it directly; otherwise, we implemented the baselines based on the descriptions in their respective papers. To ensure fair comparison, all methods were evaluated using the same common hyperparameters, such as learning rate, batch size, optimizer, initialization method, etc.

We performed a grid search on the main hyperparameters for all methods to ensure fair comparisons. We report the best results for the baselines. Specifically, the parameters we tuned include: temperature for ICC (0.05,0.1,0.5,1.0), tau for ARF and LCRON (1,20,50,100,200,1000); alpha (0,0.25,0.5,0.75,1) for RankFlow; and top-k (10,20,30,40) and smooth factor (0,0.25,0.5,0.75,1) for FS-LambdaLoss. BCE and FS-RankNet do not have independent hyperparameters.

### C.2. Experiments on Three-stage Cascade Ranking

In the main text, we only show the experiments on two-stage cascade ranking scenarios. Although LCRON is readily extended to cascade ranking systems with more than two stages, which is emphasized in Section 4.1, there is no experimental evidence for this claim. To further verify the scalability of LCRON for $T > 2$ stages, we also conduct experiments under the three-stage cascade ranking setting, which is also a typical setting for $T$ in real-world cascade ranking applications. Limited

*Table 6.* Experimental results for three-stage cascade ranking. Each method was run 5 times, and the results are reported as mean±std. ∗ indicates the best results. Bold numbers indicate that LCRON shows statistically significant improvements over the baselines, as determined by a t-test at the 5% significance level. The test set is the last day, with the remaining data used for training.

| Method/Metric | Joint | Ranking | | Pre-ranking | | Retrieval | |
|---|---|---|---|---|---|---|---|
| | Recall@10@20 ↑ | Recall@10@20 ↑ | NDCG@10 ↑ | Recall@10@30 ↑ | NDCG@10 ↑ | Recall@10@40 ↑ | NDCG@10 ↑ |
| BCE | 0.7191±0.0005 | 0.6574±0.0011 | 0.5714±0.0009 | 0.8814±0.0009 | 0.6382±0.0007 | 0.9709±0.0006∗ | 0.6350±0.0019∗ |
| ICC | 0.6386±0.0071 | 0.6196±0.0120 | 0.5794±0.0038 | 0.7682±0.0408 | 0.4925±0.0463 | 0.8526±0.0467 | 0.4754±0.0679 |
| RankFlow | 0.7308±0.0005 | 0.7230±0.0008 | 0.6400±0.0008 | 0.8729±0.0014 | 0.6396±0.0008 | 0.9611±0.0008 | 0.6265±0.0013 |
| FS-RankNet | 0.6200±0.0010 | 0.6224±0.0008 | 0.5756±0.0006 | 0.8038±0.0006 | 0.5733±0.0008 | 0.9373±0.0008 | 0.5678±0.0008 |
| FS-LambdaLoss | 0.7319±0.0038 | 0.7292±0.0042 | 0.6431±0.0014∗ | 0.8803±0.0029 | 0.6443±0.0024∗ | 0.9662±0.0012 | 0.6297±0.0022 |
| ARF | 0.7256±0.0004 | 0.7251±0.0005 | 0.5675±0.0036 | 0.8712±0.0008 | 0.5099±0.0074 | 0.9612±0.0004 | 0.4268±0.0031 |
| ARF-v2 | 0.7332±0.0020 | 0.7285±0.0051 | 0.6430±0.0015 | 0.8777±0.0064 | 0.6438±0.0031 | 0.9649±0.0029 | 0.6284±0.0039 |
| LCRON | **0.7390±0.0008**∗ | 0.7338±0.0008∗ | 0.6017±0.0009 | 0.8859±0.0007∗ | 0.6010±0.0031 | 0.9678±0.0012 | 0.5758±0.0055 |

*Table 7.* Experimental results for directly using differentiable sorting operators to learning the rank, and test LCRON with different differentiable sorting operators. The experiments are under the settings of the two-stage cascade ranking. Each method was run 5 times, and the results are reported as mean±std. ∗ indicates the best results. The test set is the last day, with the remaining data used for training.

| Method/Metric | Joint | Ranking | | Retrieval | |
|---|---|---|---|---|---|
| | Recall@10@20 ↑ | Recall@10@20 ↑ | NDCG@10 ↑ | Recall@10@30 ↑ | NDCG@10 ↑ |
| SoftSort | 0.8103±0.0013 | 0.8138±0.0011 | 0.7148±0.0006 | 0.9386±0.0003 | 0.7066±0.0005 |
| NeuralSort | 0.8210±0.0016 | 0.8233±0.0007 | 0.7138±0.0004 | 0.9469±0.0010 | 0.6979±0.0013 |
| LCRON(Softort) | 0.8723±0.0008 | 0.8720±0.0009 | 0.7246±0.0096 | 0.9703±0.0015∗ | 0.7035±0.0265 |
| LCRON(NeuralSort) | 0.8732±0.0005∗ | 0.8731±0.0004∗ | 0.7292±0.0008∗ | 0.9700±0.0004 | 0.7152±0.0009∗ |

by our industrial scenario, we can only implement two-stage experiments. Thus, we only conduct the three-stage experiments on the public benchmark. Concretely, we constructed a three-stage cascade ranking system ($T = 3$) on RecFlow (Liu et al., 2025), using its prerank_neg, coarse_neg, rank_neg, rerank_neg, and rerank_pos samples. The samples of rerank_pos are treated as ground truth. The three stages utilized DSSM (Huang et al., 2013), MLP (Rosenblatt, 1958), and DIN (Zhou et al., 2018) architectures, respectively, which are widely adopted in industrial recommendation systems for Matching, Pre-ranking, and Ranking stages. The results are shown in Table 6, formatted as mean±std. **LCRON still significantly outperforms the baselines on end-to-end recall, suggesting the scalability of LCRON to cascade ranking systems with more than two stages**.

## C.3. Experiments with Different Differentiable Sorting Operators

In the ablation study in Section 5.4, we separately validate the effects of $L_{e2e}$ and $L_{single}$. It is also curious about the performance of directly applying differentiable sorting techniques. To further validate the effectiveness of LCRON, we conduct experiments that solely use differentiable sorting techniques (i.e., aligning model predictions with label permutation matrices through CE loss), which share the same underlying rationale as FS-RankNet in making models fit complete orders. In addition, LCRON's compatibility with different differentiable sorting techniques is also curious. Thus, we further conduct experiments that test SoftSort (Prillo & Eisenschlos, 2020) as an alternative to NeuralSort (Grover et al., 2019).

All experimental results are shown in Table 7, formatted as mean±std. Each method was run five times. In Table 7, "NeuralSort" and "SoftSort" represent the results of evaluating standalone the NeuralSort (Grover et al., 2019) and SoftSort (Prillo & Eisenschlos, 2020) operators respectively. "LCRON(NeuralSort)" and "LCRON(SoftSort)" represent the results of LCRON that employs NeuralSort and SoftSort as its foundation, respectively.

The results show that **LCRON significantly outperforms directly using the differentiable sorting operators**, further validating the effectiveness of our proposed method. Another fact is that LCRON(SoftSort) also achieves significantly better performance than baselines. **These experiments verify LCRON's effectiveness and generalization capability across differentiable sorting operators, also suggesting that LCRON's effectiveness could benefit from more advanced differentiable sorting techniques**.

## C.4. Sensitivity Analysis on Hyper-parameters of LCRON

We analyze the sensitivity of LCRON to its hyperparameter $\tau$, which controls the smoothness of NeuralSort (Grover et al., 2019). Table 8 summarizes the results for different values of $\tau$. The best performance is achieved at $\tau = 50$. **Even**

*Table 8.* Sensitivity analysis results of the hyper-parameter $\tau$ of LCRON on the public benchmark, under the settings of the two-stage cascade ranking. The test set is the last day, with the remaining data used for training.

| $\tau$ | Joint | Ranking | | Retrieval | |
|---|---|---|---|---|---|
| | Recall@10@20 ↑ | Recall@10@20 ↑ | NDCG@10 ↑ | Recall@10@30 ↑ | NDCG@10 ↑ |
| 1 | 0.8701 | 0.8701 | 0.7276 | 0.9677 | 0.7071 |
| 20 | 0.8709 | 0.8708 | 0.7285 | 0.9696 | 0.7145 |
| 50 | 0.8732 | 0.8729 | 0.7291 | 0.9700 | 0.7151 |
| 100 | 0.8722 | 0.8719 | 0.7292 | 0.9707 | 0.7155 |
| 200 | 0.8721 | 0.8719 | 0.7278 | 0.9703 | 0.7156 |
| 1000 | 0.8716 | 0.8711 | 0.7285 | 0.9711 | 0.7197 |

*Table 9.* Sensitivity analysis with a learning rate of 0.001 and a batch size of 512, under the settings of the two-stage cascade ranking. The test set is the last day, with the remaining data used for training. * indicates the best results. The number in bold means that our method outperforms all the baselines on the corresponding metric.

| Method/Metric | Joint | Ranking | | Retrieval | |
|---|---|---|---|---|---|
| | Recall@10@20 ↑ | Recall@10@20 ↑ | NDCG@10 ↑ | Recall@10@30 ↑ | NDCG@10 ↑ |
| BCE | 0.8181 | 0.8000 | 0.6700 | 0.9576* | 0.6918 |
| ICC | 0.7644 | 0.7662 | 0.6648 | 0.8825 | 0.5094 |
| RankFlow | 0.8326 | 0.8272 | 0.6909 | 0.9535 | 0.6922* |
| FS-RankNet | 0.7537 | 0.7543 | 0.6546 | 0.9184 | 0.6547 |
| FS-LambdaLoss | 0.8289 | 0.8250 | 0.6878 | 0.9558 | 0.6899 |
| ARF | 0.8288 | 0.8256 | 0.6316 | 0.9515 | 0.5115 |
| ARF-v2 | 0.8302 | 0.8295 | 0.6907 | 0.9550 | 0.6838 |
| LCRON | **0.8396***  | **0.8380***  | **0.6944***  | 0.9573 | 0.6839 |

**suboptimal values of $\tau$ within a wide range (e.g., $\tau = 100$ or $\tau = 200$) yield significant improvements over the baselines, demonstrating the robustness of LCRON to hyper-parameter choices**. From the results, we observe that the performance seems to exhibit an unimodal trend with respect to $\tau$, peaking at $\tau = 50$ and gradually decreasing as $\tau$ moves away from this value. This unimodal behavior provides practical guidance for hyperparameter tuning in real-world industrial applications, suggesting that a moderate value of $\tau$ is likely to yield near-optimal performance.

### C.5. Sensitivity Analysis on Batch Size and Learning Rate

We also conduct sensitivity analysis experiments on the main hyper-parameters common to all methods, namely batch size and learning rate. Concretely, we perform the sensitivity analysis on four hyperparameter configurations: (1) learning rate=0.001 with batch size=512; (2) learning rate=0.001 with batch size=2048; (3) learning rate=0.02 with batch size=512; (4) learning rate=0.02 with batch size=2048. The experimental results are shown in Table Tables 9 to 12, respectively. It is noteworthy that these methods show different degrees of effectiveness at different learning rates and batch sizes, among which ICC is particularly sensitive to these hyperparameters, revealing that its gradient is more likely to be too sharp or vanish. Most of these methods achieved the best results under learning rate=0.02 and batch size=512. It can be seen that our method consistently achieves optimal results, demonstrating the robustness of our approach.

## D. Runtime and Space Complexity Analysis

This section presents a runtime and space complexity analysis of the discussed methods, aiming to evaluate their practicality in real-world applications.

The time and space complexity of BCE is $O(DN)$, where $N$ is the number of sampled items within a single impression and $D$ is the number of impressions in the training. RankFlow involves sequential training stages, using BCE and distillation losses (each with $O(DN)$ complexity). The serial nature of these steps introduces only a constant multiplicative factor, preserving the overall $O(DN)$ complexity. Differentiable sorting techniques such as NeuralSort and SoftSort typically have a per-impression time and space complexity of $O(N^2)$, where $N$ is the number of items sampled within a single impression. This complexity is on par with classic learning-to-rank methods like RankNet (Burges et al., 2005) and LambdaLoss (Wang et al., 2018). As a result, methods based on differentiable sorting—including ICC, FS-RankNet, FS-LambdaLoss, ARF, ARF-v2, and LCRON—also exhibit an overall time and space complexity of $O(DN^2)$.

*Table 10.* Sensitivity analysis with a learning rate of 0.001 and a batch size of 2048, under the settings of the two-stage cascade ranking. The test set is the last day, with the remaining data used for training. * indicates the best results. The number in bold means that our method outperforms all the baselines on the corresponding metric.

| Method/Metric | Joint | Ranking | | Retrieval | |
|---|---|---|---|---|---|
| | Recall@10@20 ↑ | Recall@10@20 ↑ | NDCG@10 ↑ | Recall@10@30 ↑ | NDCG@10 ↑ |
| BCE | 0.8106 | 0.7947 | 0.6629 | 0.9484 | 0.6808 |
| ICC | 0.7459 | 0.7490 | 0.6497 | 0.8754 | 0.5278 |
| RankFlow | 0.8166 | 0.8135 | 0.6857* | 0.9438 | 0.6811* |
| FS-RankNet | 0.7533 | 0.7554 | 0.6526 | 0.9111 | 0.6463 |
| FS-LambdaLoss | 0.8194 | 0.8172 | 0.6843 | 0.9467 | 0.6737 |
| ARF | 0.8174 | 0.8148 | 0.6269 | 0.9443 | 0.5039 |
| ARF-v2 | 0.8202 | 0.8193 | 0.6824 | 0.9463 | 0.6632 |
| LCRON | **0.8247*** | **0.8219*** | 0.6771 | **0.9508*** | 0.6750 |

*Table 11.* Sensitivity analysis with a learning rate of 0.02 and a batch size of 512, under the settings of the two-stage cascade ranking. The test set is the last day, with the remaining data used for training. * indicates the best results. The number in bold means that our method outperforms all the baselines on the corresponding metric.

| Method/Metric | Joint | Ranking | | Retrieval | |
|---|---|---|---|---|---|
| | Recall@10@20 ↑ | Recall@10@20 ↑ | NDCG@10 ↑ | Recall@10@30 ↑ | NDCG@10 ↑ |
| BCE | 0.8637 | 0.8522 | 0.7138 | 0.9673* | 0.7027 |
| ICC | 0.4972 | 0.5037 | 0.4151 | 0.7426 | 0.4020 |
| RankFlow | 0.7935 | 0.7983 | 0.6914 | 0.9238 | 0.6607 |
| FS-RankNet | 0.8768 | 0.8772 | 0.7409 | 0.9641 | 0.7049* |
| FS-LambdaLoss | 0.8778 | 0.8792 | 0.7430* | 0.9658 | 0.6968 |
| ARF | 0.8704 | 0.8739 | 0.6843 | 0.9583 | 0.5147 |
| ARF-v2 | 0.8776 | 0.8803 | 0.7345 | 0.9635 | 0.6935 |
| LCRON | **0.8841*** | **0.8859*** | 0.7396 | 0.9671 | 0.6969 |

In real-world applications, $N$ is usually small (e.g., 20 in our system). Moreover, the main computational cost comes from the model's prediction generation, which is independent of the specific loss function used. As a result, the computational cost from the $O(DN^2)$ operations in the loss function remains manageable. Under such settings, LCRON should incur no additional training overhead compared to baseline methods. To validate this, we conducted experiments on RecFlow using A800 GPUs and recorded the GPU memory usage and runtime. Taking two-stage cascade ranking experiments ($N$ is 40) on RecFlow as an example, the maximum of GPU memory usage for BCE, ICC, RankFlow, FS-RankNet, FS-LambdaLoss, ARF, ARF-v2, and LCRON was approximately 37.7 / 38.2 / 38.2 / 37.7 / 38.2 / 37.3 / 37.3 / 38.2 GB, respectively. The one-epoch runtimes were 5358 / 5376 / 5057 / 5104 / 5076 / 5362 / 5573 / 5418 seconds, respectively. All methods show comparable runtime performance, with minor differences likely due to runtime environmental factors rather than algorithmic complexity. In terms of GPU memory usage, model parameters dominate the consumption. Differences in space complexity among the various loss functions have a negligible impact on overall GPU memory usage.

In summary, the analysis demonstrates that LCRON does not introduce significant computational or memory overhead compared to existing methods, making it well-suited for industrial deployment.

# E. Limitations and Future Work

## E.1. Limitations of Using Soft Permutation Matrix to Express the Probability of Top-k Set Selection

For a permutation matrix $A$, the elements $A[j, i]$ and $A[k, i]$ represent the probabilities that the $i$-th element of the vector ranks $j$-th and $k$-th, respectively. For any $j \neq k$, the values of $A[j, i]$ and $A[k, i]$ are mutually exclusive; in other words, an increase in one value will suppress the other. This characteristic leads to a challenge in LCRON: although Equation 10 expresses the probability of an item appearing in the top-$q_i$ set, when optimizing the probability of the ground-truth item (assumed to be the $i$-th item) using our proposed loss, LCRON tends to simultaneously increase both $A[j, i]$ and $A[k, i]$. This results in gradient conflicts and cancellations, slowing down the optimization process and degrading performance. Moreover, this issue becomes more pronounced as $q_i$ increases.

As mentioned in the main text, to improve performance, we set $q_1$ and $q_2$ in Equation 11 to 10. While this somewhat

*Table 12.* Sensitivity analysis with a learning rate of 0.02 and a batch size of 2048, under the settings of the two-stage cascade ranking. The test set is the last day, with the remaining data used for training. * indicates the best results. The number in bold means that our method outperforms all the baselines on the corresponding metric.

| Method/Metric | Joint | Ranking | | Retrieval | |
|---|---|---|---|---|---|
| | Recall@10@20 ↑ | Recall@10@20 ↑ | NDCG@10 ↑ | Recall@10@30 ↑ | NDCG@10 ↑ |
| BCE | 0.8582 | 0.8469 | 0.7086 | 0.9691 | 0.7061 |
| ICC | 0.5061 | 0.4927 | 0.4151 | 0.7615 | 0.4647 |
| RankFlow | 0.8722 | 0.8729 | 0.7360 | 0.9625 | 0.6910 |
| FS-RankNet | 0.7884 | 0.7924 | 0.6875 | 0.9277 | 0.6630 |
| FS-LambdaLoss | 0.8726 | 0.8728 | 0.7368* | 0.9689 | 0.7141 |
| ARF | 0.8667 | 0.8680 | 0.6786 | 0.9632 | 0.5364 |
| ARF-v2 | 0.8725 | 0.8730 | 0.7323 | 0.9691 | 0.7157* |
| LCRON | **0.8785*** | **0.8785*** | 0.7327 | **0.9713*** | 0.7132 |

compromises the physical meaning of the end-to-end loss, it achieves a better trade-off between alignment with physical meaning and ease of gradient optimization. If we align $q_1$ and $q_2$ with the cascade ranking setup (i.e., setting $q_1 = 30$ and $q_2 = 20$ for the end-to-end loss), the end-to-end recall of LCRON corresponding to Table 2 would decrease to $0.8714 \pm 0.0008$.

We believe that future research should explore new differentiable operators directly designed for top-k set selection, rather than constructing the probability distribution based on sorting results. Such operators could potentially resolve the conflict issues observed in LCRON. Alternatively, another feasible approach might involve introducing carefully designed weights for different $A[j, i]$ and $A[k, i]$ terms in Equation 11, thereby mitigating the conflict problem.

### E.2. Limitations of the Fusion of Multiple Surrogate Losses

LCRON adopts UWL (Kendall et al., 2018) as a strategy to fuse different surrogate losses (i.e., $L_{e2e}$ and $L_{single}$). As shown in Table 2 and Table 4, UWL demonstrates robustness by reducing the need for manual hyperparameter tuning without compromising performance. However, it is worth noting that UWL makes an implicit assumption—originally proposed in its paper—that each loss follows a Gaussian distribution. This assumption may not hold in practice, making UWL more of a heuristic approach that primarily aims at balancing the scales of different loss terms rather than providing theoretically grounded fusion.

It remains unclear whether this strategy can maintain its effectiveness and robustness across broader scenarios, such as those involving varying model capacities or data complexities. Therefore, exploring more principled approaches for combining multiple losses in LCRON is an important direction for future work.

We argue that the problem of loss combination in LCRON can be viewed as a special case of multi-task learning, where the goal is to combine multiple objectives in a way that better optimizes the final target. Advanced techniques from the fields of multi-task learning (Liu et al., 2019; Chen et al., 2020; Wang et al., 2021) and meta-learning (Finn et al., 2017; Jamal & Qi, 2019; Lee & Yoon, 2024) are likely to offer valuable insights into improving both the effectiveness and generalization of the LCRON framework. As discussed in ARF (Wang et al., 2024), certain sub-losses may carry overlapping or hierarchical semantic meanings. Different settings—such as models with varying capacities or datasets with different levels of complexity—may benefit from distinct weighting strategies. Designing a dedicated meta-learner that adapts the loss weights based on the characteristics of the model and data might lead to a more universally robust solution.

### E.3. Computational Complexity Limitations of LCRON for Cascade Top-K Selection

LCRON utilizes a soft permutation matrix generated through differentiable sorting techniques, resulting in an $O(n^2)$ computational complexity. Although an $O(n^2)$ complexity is generally acceptable (as many mainstream learning-to-rank methods, such as LambdaLoss (Wang et al., 2018), also exhibit this complexity), it still falls short compared to the theoretical optimal complexity of $O(n)$ for hard top-k selection problems.

A key direction for future research could be to explore how to directly relax hard top-k selection methods to obtain differentiable operators for top-k selection and cascade top-k selection with a complexity of $O(n)$. Moreover, investigating how more computationally efficient algorithms can be combined with data scaling-up strategies—and their potential impact on industrial applications—is also likely to be of great significance.

