# OpenReview forum: "Learning Cascade Ranking as One Network"
_ICML.cc/2025/Conference — ICML 2025 poster_

### Official Review · Reviewer_VkMd · 2025-02-27

**Overall Recommendation:** 4

**Summary:**

The paper introduces LCRON (Learning Cascade Ranking as One Network), a novel end-to-end training framework for multi-stage ranking systems. LCRON formulates cascade ranking as a unified network with a new surrogate loss that aligns all stages with the overall top-$k$ selection objective​. In particular, it derives a differentiable lower bound on the probability that ground-truth items survive through all stages, and uses this as the primary training signal for the entire cascade.

To complement the end-to-end loss, the authors design an auxiliary per-stage loss that optimizes each stage’s Recall in isolation​. This single-stage loss (inspired by prior work on differentiable ranking) ensures each model selects ground-truth items from the full candidate set, mitigating issues like gradient vanishing and tightening the bound between the surrogate objective and true cascade performance.

Experiments demonstrate significant improvements with LCRON over existing cascade ranking methods. On the public RecFlow benchmark (a multi-stage recommendation dataset), LCRON achieves the best end-to-end Recall under realistic streaming evaluations​. In an industrial online advertising platform, LCRON delivered substantial business gains (e.g. +4.10% revenue and +1.60% user conversions compared to the strongest baseline)​. These results validate that aligning training with the cascade’s global objective and enforcing cross-stage consistency leads to a more effective and robust multi-stage ranking system​.

**Claims And Evidence:**

Yes

**Essential References Not Discussed:**

Fine-Grained Stage Alignment: One relevant recent work that was referenced but not discussed in detail is the FAA: Fine-grained Attention Alignment for cascade ranking (Li et al., 2023)​. FAA addresses the cascade ranking problem by aligning representations (attention) between stages. While LCRON approaches cascade optimization from a loss function perspective, FAA’s approach is complementary – focusing on feature consistency. A brief discussion of how LCRON’s objective alignment differs from or could be combined with representation alignment (as in FAA) would strengthen the related work section.

**Experimental Designs Or Analyses:**

The experimental evaluation is comprehensive and solid. The authors use both offline experiments (on a public dataset) and online experiments (live A/B test in an industrial system), which is a strong indicator of the method’s practical value. The offline experiments include streaming evaluations (multiple train-test splits over time)​, which adds credibility by simulating a real production training pipeline. These design choices make the results more trustworthy and less prone to overfitting on a single static split.

The paper includes analyses to isolate the impact of each component. Notably, an ablation study removes the end-to-end loss or the single-stage losses to show their contributions. The results (as described in the text) highlight that without $L_\text{single}$ or without $L_{e2e}$, the performance drops, confirming that both are needed for the full benefit. This analysis supports the claim that the combination of losses is effective. There’s also mention of a sensitivity analysis (in Appendix C) for hyperparameters, indicating the authors checked robustness of the results with respect to loss weighting or other parameters – a good experimental practice.

One aspect not fully explored is the effect of cascade depth. The experiments seem to focus on a two-stage cascade (likely because RecFlow has two ranked stages). It would strengthen the paper to see an experiment with $M=3$ stages or more, to verify that LCRON’s benefits carry over to longer cascades. For example, adding an extra intermediate ranking stage and evaluating whether LCRON still outperforms baselines (and how the losses scale) would address any concern that the method is tuned specifically to two stages. This is a missing analysis that would be valuable, although its absence does not invalidate the current results – it’s more about demonstrating scalability.

**Methods And Evaluation Criteria:**

Yes

**Other Comments Or Suggestions:**

1. Typos/Grammar: There are a few minor typos that should be corrected for the camera-ready version. For example, the abbreviation explanation for LCRON in the introduction reads “Learnig Cascade Ranking as One Network”​, missing an “n” in “Learning.” Such small errors should be fixed for clarity. Also, “auxillary” should be “auxiliary” (noticed in a section heading or text around L1095-L1103​).

2. The notation $(P_{\pi})\_i$​ is a bit confusing – it looks like a stray parenthesis. It would be clearer to denote this as $P_{\pi}(i)$ or something like $P_{\pi_i}$ if the intent is to index the vector $P_{\pi}$. Additionally, in Equation 10 (the definition of $P_{M_i}^{q_i}$ via the soft permutation matrix), it might be clearer to denote the resulting probability as $\hat{P}\_{M_i}^{q_i}$since it’s obtained through an approximation (soft sorting). Consistently using the hat notation for approximated probabilities (as done for $\hat{P}_{CS}$) would help the reader keep track of what is exact vs. relaxed.

**Other Strengths And Weaknesses:**

Strengths:

The idea of training all cascade stages as one unified network with a bound-approximating loss is a notable innovation. While it builds on elements from prior work (full-stage sampling, differentiable sorting), the particular combination – especially the novel end-to-end loss formulation ($L_\text{e2e}$) and the bound-tightening strategy with $L_\text{e2e}$ – is original. This approach has not been explicitly done before in the literature, making it a fresh contribution.

The contributions have high significance for both research and industry. Improving multi-stage ranking has direct implications for large-scale recommender systems and search engines. The fact that LCRON showed measurable gains in a production environment (revenue and conversions) indicates that this method can impact real systems, not just benchmark scores. For the research community, LCRON’s approach could inspire more work on global objective optimization in cascades and on using differentiable ranking techniques in multi-stage pipelines. It effectively addresses a gap in the literature, so its acceptance would add valuable knowledge and potentially a new baseline for others to compare against.

Weaknesses:

One weakness is the complexity introduced by the method. Training with differentiable sorting and multiple loss components (even if combined via learned weights) can be harder to implement and tune than traditional methods. The paper mitigates this by using the UWL scheme to automatically balance loss weights, but the approach still requires careful engineering (e.g., setting the softmax temperature $\tau$ for NeuralSort). Another minor weakness is that some claims (like robustness across model capacities) were not directly verified, as noted earlier. Lastly, the method’s benefit was clearly shown for two-stage cascades; it’s an open question how it performs with more stages or in scenarios with dramatically different stage characteristics (this could be explored in future work). These weaknesses, however, are not fundamental flaws but areas to keep in mind when applying the method.

**Questions For Authors:**

1. In the abstract, you write “design an auxiliary loss for each stage to drive the reduction of this bound”​. Could you clarify this phrasing? Does it mean that optimizing the auxiliary loss empirically tightens the lower bound (i.e., increases $\hat{P}\_{CS}$ toward $P_{CS}$)? Please elaborate on how $L_\text{single}$ concretely contributes to reducing the gap between the lower bound and the true joint probability – for instance, is there a way to measure or prove this reduction happens as $L_\text{single}$ is minimized? A bit more intuition here would help: we see the math in Appendix A, but a clearer explanation of how the auxiliary loss “drives the bound’s reduction” would be appreciated.

2. Notation $(P_{\pi})\_i$ – is this a typo? In Section 4.2, you define $P_{\pi}$ as the probability of a sampling $\pi$, and then use the notation $(P_{\pi})\_i$​. This notation was a bit hard to parse – it looks like $P_{\pi}$ might be a vector or distribution, and you’re referring to its $i$-th component. Could you confirm what $(P_{\pi})\_i$ means exactly? If $P_{\pi}$ is a distribution over sampled sets, perhaps $(P_{\pi})\_i$ is the probability that a particular item $i$ is included? The parentheses made it read like a possible typo. Should it be $P{\pi_i}$ (the probability of a specific permutation) or $P(\pi_i)$? Clarifying this would help readers follow the derivation in Eq. 6 without confusion.

3. Have you considered or tested LCRON on a cascade with three stages (or more)? While RecFlow provides two-stage data, it would be insightful to know how the approach scales to an additional stage. For example, if we had a retrieval model $M_1$, an intermediate ranker $M_2$, and a final ranker $M_3$, would the LCRON formulation easily extend (with $L_\text{e2e}$ using $\prod_{i=1}^{3} P^{q_i}\_{M_i}$ and each stage having $L_\text{single}$)? If you have any preliminary results or reasoning for $M=3$, please share them. This would help convince readers (and practitioners) that LCRON generalizes beyond the two-stage scenario. Are there any expected difficulties for $M>2$ (e.g., increased gradient variance or more hyperparameters), or does it plug in seamlessly?

**Relation To Broader Scientific Literature:**

This paper positions itself relative to three main recent approaches in multi-stage ranking: RankFlow (Qin et al., 2022), FS-LTR (Zheng et al., 2024), and ARF (Wang et al., 2024). RankFlow introduced joint optimization via iterative feedback between stages, and FS-LTR proposed training all stages on the full cascade data (to mitigate sample bias)​. However, neither explicitly optimizes the final-stage metric. ARF introduced a differentiable surrogate loss for Recall but only for a single stage​. LCRON clearly extends these ideas: it combines the full-cascade training philosophy of FS-LTR with the metric-driven loss of ARF, achieving an end-to-end training that RankFlow attempted, but in a single unified model rather than an iterative process. This represents a notable advancement, as no existing approach simultaneously addressed both the sample bias and objective misalignment before (as the authors point out)​.

The cascade ranking concept isn’t new (e.g., Wang et al., 2011 introduced an efficient cascade ranker; Gallagher et al., 2019 (ICC) optimized fused stage scores via LambdaRank​). LCRON differs by focusing on directly optimizing the selection probability of relevant items. Earlier methods often optimized proxy objectives (like combined scores or separate stage objectives) and could suffer from stage inconsistency or bias. By using differentiable sorting and a probabilistic formulation, LCRON provides a more principled end-to-end solution. This is a meaningful improvement on the foundations laid by those works, aligning with a broader trend in ranking research to move from heuristic multi-stage training to theoretically grounded joint optimization.

Compared to FS-LTR (which is a strong recent baseline), LCRON adds the missing piece of objective alignment. FS-LTR trained all stages together but still used traditional loss functions per stage, whereas LCRON introduces losses that correspond to Recall directly​. Similarly, compared to ARF’s single-stage recall optimization, LCRON shows how to incorporate that idea across an entire cascade and handle the interactions between stages (for example, by multiplying probabilities from stage 1 and stage 2). The results in the paper demonstrate that these improvements are not just theoretical – LCRON outperforms ARF and FS-LTR in practice, indicating the combination of techniques is effective.

**Theoretical Claims:**

The paper’s theoretical core is the derivation of a lower bound for the joint survival probability of ground-truth items in the cascade. This is presented by defining an approximate joint probability $\hat{P}\_{CS}$ as the product of per-stage selection probabilities​ and showing that $\hat{P}\_{CS}$ is a provable lower bound of the true cascade selection probability $P_{CS}$​. The steps in this derivation (Eq. 7–8 in the paper) appear to be sound and mathematically correct, relying on the fact that certain fractions are always $\leq 1$. The bound is logically valid given the independence assumption between stages’ selection events.

The authors claim that the auxiliary single-stage loss “tightens the bound” – i.e., reduces the gap between the lower bound $\hat{P}\_{CS}$ and the true $P_{CS}$. In Appendix A, they analyze the gap $\Delta = P_{CS} - \hat{P}\_{CS}$ and relate it to the consistency between stages​. The theory suggests that if each stage more reliably retrieves the relevant item (as enforced by $L_\text{single}$ on the full candidate set), then $\hat{P}\_{CS}$ approaches $P_{CS}$. This reasoning is sound in principle – a more consistent cascade (each stage capturing the true item) will make the product approximation more accurate. The provided equations support this claim, though they rely on expectations over permutations and some assumptions of stage-independence.

One minor concern is that the paper could better explain in words some of the theoretical conclusions. For instance, the phrase “drive the reduction of this bound”​ is used to describe the effect of the auxiliary loss – while the math shows $L_{single}$ helps close the gap, a clearer intuitive explanation of how reducing the bound gap translates to better Recall would help. Additionally, the notation $\left(P_{\pi}\right)\_i$ introduced in the theoretical section is a bit confusing​. It represents the $i$th component of the sampling distribution $P_{\pi}$, but it initially looks like a stray parenthesis. This could be clarified to avoid any doubt in the correctness of the notation (see questions for authors). Overall, the theoretical content is sound; it supports the method’s design well, with just a few places where more explicit justification or clearer notation would strengthen confidence.

---

> ### Author Rebuttal · Authors · 2025-03-31
>
> Thank you for your detailed and insightful review.
>
> For the questions:
>
> 1) Yes, the abstract intends to state that L_single helps reduce the gap between $P_{CS}$ and $\hat{P_{CS}}$. In lines 576-588 of the appendix, we explained the conditions for the gap to be reduced to 0 (i.e., the equation in eq14 holds). It can be found that the consistency of the top q2 sets selected by the two models is a sufficient condition for the gap reducing to 0, so optimizing L_single helps to reduce the gap. Indeed, if the role of auxiliary loss can be explained more intuitively in the abstract and introduction, it will make the article easier to understand. Thank you very much for the suggestion regarding the description of the abstract. However, it is a little difficult to intuitively explain why l_single works without a mathematical description. We welcome suggestions for clearer phrasing and will revise accordingly.
>
> 2) Thank you very much for your careful evaluation of our work. There should be $P_{\pi}$ rather than ${(P_{\pi})\_i}$, which is indeed a typo and leads to confusion. $P_{\pi}$ represents the probability distribution of sampling to set $\pi$.
> When $\pi$ is given, $P_{\pi}$ is a scalar. In addition, to make the description more rigorous, all $\pi\sim P_{M_1}^{q_1}$ in the text should be replaced by $\pi\sim P_{\pi}$, indicating that $\pi$ is sampled from $P_{\pi}$.
> $P_{M_1}^{q_1}$ only represents the mean vector of the sampling distribution. We will fix these typos.
>
> 3) Limited by our industrial scenario, we can only implement two-stage experiments. Thanks for your recognition of the comprehensive and solid experimental part. We also believe that the experiment in the scenario of M>2 can further verify the scalability of LCRON and enhance the depth of this paper. RecFlow contains data from more stages. We constructed a three-stage (M=3) cascade ranking system on the public RecFlow benchmark, using its prerank_neg, coarse_neg, rank_neg, rerank_neg, and rerank_pos samples (rerank_pos is the ground truth). The three stages utilized DSSM, MLP, and DIN architectures, respectively. The results are shown in the following, formatted as mean±std(p-value). LCRON still outperforms the baselines (statistically significant), showing the scalability of LCRON.
>
> |Method|Joint Recall|
> |--|--|
> |BCE|0.7191±0.0005(0.0000)|
> |ICC|0.6386±0.0071(0.0000)|
> |RankFlow|0.7308±0.0005(0.0000)|
> |FS-RankNet|0.6200±0.0010(0.0000)|
> |FS-LambdaLoss|0.7319±0.0038(0.0214)|
> |ARF|0.7256±0.0004(0.0000)|
> |ARF-v2|0.7332±0.0020 (0.0076)|
> |LCRON(ours)|**0.7390±0.0008**|
>
> For the weaknesses:
>
> 1) About Complexity & Deployment:
> Differentiable sorting techniques such as NeuralSort and SoftSort typically have an O(n²) complexity, which is the same as common LTR methods like LambdaLoss, where n refers to the number of sampled items within a single impression. In real-world applications, n is usually not too large (e.g. n=20 in our system), LCRON incurs no additional training cost compared to the baselines.
> For public experiments, we use A800 GPU to conduct public experiments. The GPU memory used for BCE, ICC, RankFlow, FS-RankNet, FS-LambdaLoss, ARF, LCRON are 28.4/28.9/28.9/28.4/28.9/28.4/28.9GB, respectively. The runtime of one epoch of them are 5358/5376/5057/5104/5076/6145/5339s, respectively.
> Deployment requires training stages in a single TensorFlow job and loading weights into separate meta files—it is not difficult for industrial serving teams (Appendix Lines 643–646). Moreover, $\tau$ is the sole hyper-parameter of LCRON, thus its use typically requires only a small amount of hyperparameter tuning. To sum up, LCRON incurs no significant training/deployment overhead in real-world applications.
>
> 2) Scalability: Please see the response to Q3 above.
>
> 3) Regarding the robustness across capacities: Our original claim intended to highlight LCRON's consistent performance across different model architectures (e.g., DSSM+MLP, DSSM+DIN), suggesting robustness on capacities. We agree that the experiments do not directly validate robustness across model capacities. To address this, we will refine this claim of the abstract in the revised version and add a discussion in the Limitations and Future Work section to guide further exploration.
>
> For the missing reference:
> Thank you for pointing out this work. We will discuss FAA in the Related Work section to enhance the comprehensiveness of related work.
>
> For the typos:
> Thank you for pointing out these typos. Additionally, we noticed that the Recall and NDCG for each single stage in the manuscript were mistakenly swapped. All noted errors will be corrected, and we will thoroughly proofread the manuscript to ensure accuracy.
>
> Due to space limitations, we only show key results in the rebuttal text. Full additional results can be found in this anonymized github link: https://anonymous.4open.science/r/2025038594/
>
> If you have any further questions or concerns, we will make every effort to provide further clarification.

---

### Official Review · Reviewer_abHQ · 2025-03-08

**Overall Recommendation:** 3

**Summary:**

This paper proposes LCRON (Learning Cascade Ranking as One Network), a new method for optimizing cascading sorting systems. LCRON implements end-to-end training through two agent loss functions (Le2e and Lsingle) to ensure that the goals of each stage are consistent with the overall goals of the system and enhance the interaction between stages. Experiments have shown that LCRON outperforms existing methods in both public benchmarks and industrial applications, significantly improving ad revenue and user conversion rates.

## update after rebuttal
Thanks for your careful response, and I consider the previous score reasonable and will keep the previous rating.

**Claims And Evidence:**

The paper's main claims are supported by convincing evidence. Extensive experiments on four datasets demonstrate LCRON’s superiority over state-of-the-art methods across multiple metrics.

**Essential References Not Discussed:**

There are no related works that are not currently discussed in the paper.

**Experimental Designs Or Analyses:**

I checked the validity of the experimental designs and analyses. The experiments are conducted on the RecFlow benchmark dataset, which is specifically designed for cascade ranking systems and includes multi-stage samples from real-world recommendation systems. The results are evaluated using the Recall@K@m metric, which is a standard evaluation criterion for cascade ranking systems. Additionally, the paper includes an ablation study, streaming evaluation, and online A/B testing to comprehensively validate the effectiveness of LCRON. The issues are listed behind in the Weaknesses.

**Methods And Evaluation Criteria:**

The proposed methods make sense for the problem. LCRON innovatively introduces two novel surrogate loss functions (Le2e and Lsingle) to align the training objectives across multiple stages of cascade ranking, ensuring end-to-end optimization and enhanced interaction awareness between stages. The experimental evaluation utilizes the RecFlow benchmark dataset, which is specifically designed for cascade ranking systems and includes multi-stage samples from real-world recommendation systems. Additionally, the evaluation compares LCRON against state-of-the-art baseline methods, demonstrating its effectiveness in both public benchmarks and industrial applications.

**Other Comments Or Suggestions:**

I would like to learn about the authors' response to the weaknesses listed above, which may give me a clearer perspective on the paper's contribution.

**Other Strengths And Weaknesses:**

The paper proposes LCRON (Learning Cascade Ranking as One Network), a framework for optimizing cascade ranking systems by introducing two new surrogate loss functions (Le2e and Lsingle) to align training objectives across stages and enable end-to-end optimization. This approach is original and significant, as it addresses key limitations in traditional cascade ranking training methods, such as misaligned objectives and insufficient stage interaction. The application to real-world recommendation and advertising systems further highlights its practical significance.
Weaknesses:
1. Comparison with Existing Techniques: While LCRON is compared with state-of-the-art methods like RankFlow and FS-LTR, a deeper comparison with other differentiable ranking techniques or multi-stage optimization approaches could further highlight its advantages.
2. Ablation Studies: Although the paper includes an ablation study, additional experiments to isolate the impact of individual components (e.g., the role of differentiable sorting or the interaction between Le2e and Lsingle) would strengthen the claims.

**Questions For Authors:**

I would like to learn about the authors' response to the weaknesses listed above, which may give me a clearer perspective on the paper's contribution.

**Relation To Broader Scientific Literature:**

LCRON advances cascade ranking by addressing limitations of traditional methods. It builds on interaction-aware training (e.g., RankFlow, FS-LTR) and differentiable sorting (e.g., ARF), introducing novel losses (Le2e, Lsingle) for end-to-end optimization and stage-specific supervision. These contributions align with broader trends in multi-task learning and differentiable techniques, offering a robust solution for cascade ranking systems.

**Theoretical Claims:**

I checked the correctness of the proofs for theoretical claims, including the derivation of the lower bound for the survival probability of ground-truth items in the cascade ranking system.  The theoretical analysis is sound and aligns with the experimental results, showing that Lsingle effectively reduces the gap and enhances the overall performance of the cascade ranking system. No significant issues were found in the theoretical claims.

---

> ### Author Rebuttal · Authors · 2025-03-31
>
> Thank you very much for your detailed and insightful review.
>
> For weaknesses 1 & 2:
>
> To the best of our knowledge, we have already compared LCRON with existing multi-stage optimization methods. **To further validate the effectiveness of solely using differentiable sorting techniques (i.e., aligning model predictions with label permutation matrices through CE loss)**, which shares the same underlying rationale as FS-RankNet in making models fit complete orders, **we add two new baselines**. We conduct experiments on NeuralSort [1] and SoftSort [2].
> In ablation studies, we separately validated the effects of $L_{e2e}$​ and $L_{single}$. Since differentiable sorting techniques form the foundation of both $L_{e2e}$ and $L_{single}$, the results of **"NeuralSort"** and **"SoftSort"**, which evaluate standalone differentiable sorting techniques, **also serve as an ablation of LCRON**. Moreover, **we further conduct experiments using the SoftSort [2] operator as the foundation of LCRON, to study its generalization capability across different differentiable sorting operators**. All additional results are shown in the following table, formatted as mean±std(p-value). Each method was run 5 times, and we conducted t-tests between LCRON (NeuralSort) and each of the other methods.
>
> |          Methods          |    Joint Recall (Golden Metric)     |   Recall of Ranking Model    | NDCG of Ranking Model  |  Recall of Retrieval Model   |   NDCG of Retrieval Model    |
> | ------------------------- | ----------------------------------- | ---------------------------- | ---------------------- | ---------------------------- | -----------------------------|
> | **NeuralSort** |        0.8210±0.0016(0.0000)        |    0.8233±0.0007(0.0000)     | 0.7138±0.0004(0.0000)  |    0.9469±0.0010(0.0000)     |    0.6979±0.0013(0.0000)     |
> | **SoftSort**              |        0.8103±0.0013(0.0000)        |    0.8138±0.0011(0.0000)     | 0.7148±0.0006(0.0000)  |    0.9386±0.0003(0.0000)     |    0.7066±0.0005(0.0000)     |
> | **LCRON(SoftSort)**       |        0.8723±0.0008(0.0615)        |    0.8720±0.0009(0.0485)     | 0.7246±0.0096(0.3395)  |    0.9703±0.0015(0.6750)     |    0.7035±0.0265(0.3782)     |
> | **LCRON(NeuralSort)**     |            0.8732±0.0005            |        0.8731±0.0004         |     0.7292±0.0008      |        0.9700±0.0004         |        0.7152±0.0009         |
>
> The results show that LCRON(SoftSort) can also achieve significantly better performance than baselines. These experiments verify LCRON's generalization capability across different differentiable sorting operators, also suggesting that LCRON's effectiveness could benefit from more advanced differentiable sorting techniques.
>
> We will add these results to the new version. If you have any further questions or concerns, we will make every effort to provide further clarification.
>
> References:
>
> [1] Grover, A., Wang, E., Zweig, A., and Ermon, S. Stochastic optimization of sorting networks via continuous relaxations. In ICLR, 2019.
>
> [2] Prillo, S. and Eisenschlos, J. Softsort: A continuous relaxation for the argsort operator. In International Conference on Machine Learning, pp. 7793–7802. PMLR, 2020.

---

### Official Review · Reviewer_VSAL · 2025-03-19

**Overall Recommendation:** 3

**Summary:**

The paper addresses two key challenges in cascade ranking:
(i) the misalignment of training objectives across different stages and
(ii) the discrepancy between training and test environments caused by multi-stage ranking and filtering.

To overcome these issues, the authors propose a novel loss function comprising: an end-to-end term that optimizes the global objective of the cascade ranking system, and stage-wise terms that help models adapt to changes in the sample space distribution introduced by upstream ranking stages. The paper adopts the model architecture for all stages from RecFlow [1], focusing its contributions on refining the training loss.

[1] RecFlow, ICLR'25

**Claims And Evidence:**

The paper introduces LCRON, a novel loss function for cascade ranking that simultaneously addresses two key challenges: misalignment of training objectives across stages and discrepancies between training and testing environments. By incorporating an end-to-end loss term alongside stage-wise losses, the method aims to improve coordination across ranking stages and enhance overall system performance.

Experimental results indicate that LCRON achieves the highest end-to-end recall among the tested methods. However, a key limitation of the evaluation is that all baseline methods share the same model architecture and differ only in their loss functions. While this ensures a controlled comparison, it leaves open the question of whether the improvements stem from the proposed loss function itself or from interactions with the underlying architecture. Additionally, the most competitive baseline, FS-LambdaLoss, outperforms LCRON in both ranking and retrieval stages in terms of Recall, suggesting that while LCRON improves joint performance, its per-stage effectiveness varies. Furthermore, the overall gains in end-to-end recall, though positive, are relatively small, and validating them with statistical significance tests would strengthen the claims.

**Essential References Not Discussed:**

The paper discusses key references related to cascade ranking and provides a solid foundation for its contributions. But it should be noted that, some closely related work on the interaction between retrieval and ranking stages, such as Stochastic Retrieval-Conditioned Reranking (Zamani et al., ICTIR'22), is not covered. Including such references could provide additional context and help position the proposed approach within the broader landscape of retrieval and ranking research.

**Experimental Designs Or Analyses:**

Please see "Claims And Evidence" and "Methods And Evaluation Criteria"

**Methods And Evaluation Criteria:**

The benchmark datasets used in this paper are well-suited to the cascade ranking problem, providing a realistic testbed for evaluating multi-stage ranking systems. However, the authors limit their evaluation to a two-stage setup, even though the RecFlow benchmark includes four stages. While the approach can theoretically be extended to more stages, it remains unclear how the proposed surrogate loss function performs across all stages of a fully deployed cascade ranking system. Evaluating LCRON in a true multi-stage setting would provide deeper insights into its effectiveness and scalability.

**Other Comments Or Suggestions:**

--

**Other Strengths And Weaknesses:**

Strengths:
- The paper identifies key challenges in cascade ranking - misalignment of training objectives across stages and discrepancies between training and test environments; and introduces a novel loss function which optimizes both end-to-end performance and stage-wise alignment through a novel surrogate loss function.

- The method is evaluated on both public benchmark and an industrial dataset, with online A/B testing showing a 4.1% increase in revenue and a 1.6% increase in user conversions, highlighting its practical impact.

Weaknesses:

- While the RecFlow dataset includes four ranking stages, the paper evaluates LCRON only on a two-stage setup, leaving it unclear how well the method scales to fully deployed multi-stage cascade ranking systems.

- All baselines share the same model architecture and differ only in their loss functions, limiting the scope of comparison, and the strongest baseline (FS-LambdaLoss) outperforms LCRON in both retrieval and ranking stages in terms of Recall, raising questions about whether the overall gains justify the added complexity.

- The reported improvements in end-to-end recall are relatively small, and the paper does not provide statistical significance tests to confirm their robustness, making it difficult to determine whether the gains are meaningful or within the margin of variance.

**Questions For Authors:**

Please see above sections

**Relation To Broader Scientific Literature:**

The paper brings up two important challenges in cascade ranking and suggests using surrogate loss functions to tackle them. Since cascade ranking is widely used in industrial recommendation systems, this is a valuable contribution to the field. However, the approach would be more convincing with a stronger set of experiments, especially testing across more stages, broader set of architectures, and validating the improvements with significance tests.

**Theoretical Claims:**

The paper provides a clear explanation of the proposed loss function. However, I am unable to verify the theoretical claims due to my limited expertise in extensive mathematical proofs.

---

> ### Author Rebuttal · Authors · 2025-03-31
>
> Thanks very much for the detailed and insightful review.
>
> For the weaknesses:
>
> 1) We adopted a two-stage setup (retrieval + ranking) because two-stage cascading represents the most classic form of cascade ranking, as seen in previous works like FS-LTR. From a practical perspective, to the best of our knowledge, real-world online cascade ranking systems typically employ 3–4 stages to achieve an optimal trade-off between effectiveness and efficiency. Due to real-world constraints (e.g., business requirements, team organizations), retrieval + preranking often serves as the most feasible scenario for validating and implementing multi-stage joint optimization. Thus, we believe our experiments retain generality and sufficiently demonstrate the value of our method in real-world applications. Nevertheless, we acknowledge that testing LCRON on scenarios with >2 stages could further validate its scalability and enhance the depth of this work. Due to some limitations of our industrial scenario, we can not deploy 3 or more stages experiments. We constructed a three-stage cascade ranking system and verified the effectiveness of LCRON on RecFlow. **The detailed settings and results are in the response to reviewer VkMd. Experimental results show that LCRON still significantly outperforms baselines in this three-stage setting**.
>
> 2) **a)** We **aligned all baseline model architectures and isolated parameters across stages** strictly (as illustrated in lines 301-304). This **ensures performance improvements are not attributed to parameter sharing or architectural interactions**. Our public experiments used DSSM and DIN (with attention), while online experiments used DSSM and MLP—**both setups cover mainstream architectures (DSSM for Retrieval, MLP for Pre-ranking, attention-based models [e.g., DIN] for Ranking) in recommendation/advertising cascade systems**. We believe this sufficiently demonstrates LCRON’s generalization across architectures.
> **b)** In Table 2, the metrics under "ranking" and "retrieval" reflect individual model performance on full samples, not their combined cascade performance. **The comparison between FS-LambdaLoss and LCRON highlights that neglecting inter-stage interactions may lead to suboptimal cascade results despite strong standalone model performance**. This confirms that LCRON priorities end-to-end cascade effectiveness over individual model optimization. Note that end-to-end recall is the golden metric for cascade ranking, as illustrated in lines 314-322. **c) About the complexity, please see the response to reviewer VkMd (for weakness 1)**.
>
> 3) We prioritized significance testing (via unpaired t-tests with 5 runs per method) for ablation studies (Table 3), where smaller performance gaps required rigorous validation. The larger margins in baseline comparisons (Table 2) inherently imply statistical significance. Furthermore, online experiments confirmed LCRON’s statistically significant superiority over strong baselines like FS-LTR and ARF-v2 in public benchmarks. We believe these results collectively justify the robustness of Table 2’s conclusions. However, to address potential concerns about the statistical significance of improvements in Table 2, we conducted additional significance tests on the baselines. **The results are shown in the following table, formatted as mean±std(p-value). LCRON demonstrates statistically significant (p-value<0.05) superiority over all baselines on joint recall (namely end-to-end recall)**.
>
> |Method/Metric|JointRecall@10@20↑|
> |-------------|-------------------|
> |BCE|0.8539±0.0006(0.0000)|
> |ICC|0.8132±0.0003(0.0000)|
> |RankFlow|0.8647±0.0007(0.0000)|
> |FS-RankNet|0.7881±0.0007(0.0000)|
> |FS-LambdaLoss|0.8666±0.0016(0.0004)|
> |ARF|0.8608±0.0006(0.0000)|
> |ARF-v2|0.8678±0.0009(0.0000)|
> |LCRON(ours)|0.8732±0.0005|
>
> For the missing reference:
>
> Thank you for highlighting this work. We notice that it focuses on improving "retrieval + ranking" cascade systems but employs a non-learnable retrieval component (BM25) paired with BERT for ranking. Specifically, it jointly optimizes the number of retrieved documents N and the ranking model. This differs from our focus on joint learning across fully learnable cascade stages. We will discuss this work in the Related Work section to clarify its distinctions from our approach.
>
> &nbsp;
>
> Due to space limitations, we only show the key results (e.g., end-to-end recall) in the rebuttal text. Full additional results can be found in this anonymized github link: https://anonymous.4open.science/r/2025038594/
>
> If you have any further questions or concerns, we will make every effort to provide further clarification.

---

### Official Review · Reviewer_yid9 · 2025-03-19

**Overall Recommendation:** 4

**Summary:**

The paper "Learning Cascade Ranking as One Network" introduces LCRON, a novel approach for training cascade ranking systems in an end-to-end manner. Traditional cascade ranking architectures suffer from misalignment between training objectives across different stages and discrepancies between training and testing environments. The paper proposes a new surrogate loss function that optimizes the lower bound of the survival probability of ground-truth items through all stages, ensuring a better alignment of training objectives. The authors also introduce an auxiliary loss for each stage to improve robustness. Experimental results on public (RecFlow) and industrial benchmarks demonstrate that LCRON outperforms existing approaches in terms of recall and conversion metrics, achieving a 4.1% increase in advertising revenue and a 1.6% increase in user conversions in a real-world deployment.

**Claims And Evidence:**

The main claims of the paper are:
- LCRON aligns training objectives across all cascade ranking stages: Supported by the proposed surrogate loss function, which explicitly optimizes the recall of the entire system rather than individual stages.
- LCRON improves end-to-end recall compared to existing methods: Empirical results from RecFlow and industrial benchmarks confirm higher recall scores.
- LCRON enhances commercial performance in real-world applications: A/B testing in a real-world advertising system shows notable revenue and conversion improvements.

While these claims are mostly well-supported, the empirical results focus primarily on recall metrics, and it would be useful to evaluate additional ranking quality metrics (e.g., precision, diversity).

**Essential References Not Discussed:**

While the paper covers major references, it could benefit from discussing additional works on differentiable sorting and ranking, such as:
- Blondel, M., Teboul, O., Berthet, Q., & Djolonga, J. (2020, November). Fast differentiable sorting and ranking. In International Conference on Machine Learning (pp. 950-959). PMLR.
- Pobrotyn, P., & Białobrzeski, R. (2021). Neuralndcg: Direct optimisation of a ranking metric via differentiable relaxation of sorting. arXiv preprint arXiv:2102.07831.
- Cuturi, M., Teboul, O., & Vert, J. P. (2019). Differentiable ranking and sorting using optimal transport. Advances in neural information processing systems, 32.
- Thonet, T., Cinar, Y. G., Gaussier, E., Li, M., & Renders, J. M. (2022, June). Listwise learning to rank based on approximate rank indicators. In Proceedings of the AAAI Conference on Artificial Intelligence (Vol. 36, No. 8, pp. 8494-8502).

**Experimental Designs Or Analyses:**

The experimental design is strong and well-structured, with:
- Comparisons against state-of-the-art baselines (BCE, ICC, RankFlow, FS-LTR, ARF)
- Public and industrial benchmarks
- Ablation studies to test the contribution of each component
- Streaming evaluation to simulate real-world training conditions

A few areas for improvement are (i) Computational cost analysis: how does LCRON’s training time compare to existing methods?
(ii) Robustness to dataset shifts: given that RecFlow spans multiple time periods, an explicit analysis of temporal generalization would be beneficial.

**Methods And Evaluation Criteria:**

The paper presents a clear research question: How can cascade ranking be trained end-to-end while aligning training objectives across all stages? The hypothesis, which proposes that a surrogate loss can improve ranking alignment and recall, is consistent with the methodology and results. The experimental design is appropriate for addressing this question.

*Baselines*. The paper compares LCRON against state-of-the-art methods such as BCE, ICC, RankFlow, FS-LTR, and ARF. The chosen baselines are well-justified, covering both simple (BCE) and advanced (RankFlow, FS-LTR) approaches. However, it is unclear whether baseline results were obtained from previous papers or rerun under the same conditions. Explicit clarification on this would improve reproducibility.

*Evaluation Metrics*. The primary evaluation metric is Recall@K@M, which is relevant for cascade ranking. The authors also report NDCG@K as a secondary metric. While these metrics align with the research goal, additional discussion on trade-offs between precision, recall, and ranking diversity would be beneficial.

*Data Collection and Preprocessing*. The dataset choice (RecFlow) is appropriate, as it contains multi-stage ranking samples.
Data preprocessing steps (e.g., filtering of interactions) are not detailed. If any pruning was performed, the justification should be included.

*Data-Splitting and Generalization*. The train-test split is conducted over time, which is standard for ranking models.
All models appear to be trained on the same splits, but cross-validation techniques are not explicitly mentioned.

*Hyperparameter Optimization*. The optimization strategy is briefly discussed but lacks detail on parameter ranges and tuning procedures. It is unclear how many configurations were tested or how hyperparameters were selected, which could impact result reproducibility.

*Experiment Execution and Sensitivity Analysis*. The experimental setup appears fair, but hardware details (e.g., GPU models, memory) are missing. There is no explicit discussion of statistical significance tests (e.g., p-values or confidence intervals) in Table 2, which would strengthen the analysis.
Sensitivity analysis is limited to the temperature parameter in differentiable sorting, but other key hyperparameters (e.g., learning rate, batch size) are not explored.

**Other Comments Or Suggestions:**

- Consider including a formal convergence proof for LCRON's optimization process to demonstrate that it will always converge. Additionally, it would be helpful to analyze the conditions under which the optimization may fail or become unstable.

- It would strengthen the paper to provide formal generalization bounds for LCRON, especially in highly dynamic ranking environments, to guarantee that the model will generalize well to unseen data, beyond the empirical results.

**Other Strengths And Weaknesses:**

Other Strengths:
- Practical Relevance: The method has clear real-world applicability in advertising and recommender systems.

- Clear Paper Structure: The organization of the paper makes it easy to follow.

- Empirical Strength: The industrial deployment and A/B testing strengthen the validity of claims.

Other Weaknesses:
- Computational Complexity Not Analyzed: The additional cost of using differentiable sorting techniques is not explicitly measured, which could impact scalability in large-scale applications

**Questions For Authors:**

1. why didn't you evaluate additional ranking quality metrics (e.g., precision, diversity)?
2. were baseline results obtained from previous papers or rerun under the same conditions?
3. can you detail  which data preprocessing steps (e.g., filtering of interactions) were performed? If any pruning was performed, the justification should be included.
4. What strategy was used for hyperparameter tuning and what were the specific ranges considered for key hyperparameters? How many configurations were tested during tuning, and what criteria were used to select the final hyperparameters? Were all baselines tuned under the same conditions to ensure fair comparisons?
5. Can you specify the computational resources used for training, such as GPU models, memory, and runtime per experiment?
6. Why are statistical significance tests (e.g., p-values, confidence intervals) missing from Table 2? Can you provide evidence to confirm the robustness of the reported improvements?

**Relation To Broader Scientific Literature:**

The paper builds upon and improves several prior works in cascade ranking, particularly: LambdaRank and Learning-to-Rank approaches, Differentiable sorting techniques (NeuralSort, SoftSort), Interaction-aware training methods (RankFlow, FS-LTR).

The work is well-grounded in prior literature.

**Theoretical Claims:**

The theoretical justification for LCRON’s surrogate loss function appears sound. The derivation of the lower bound on the survival probability of ground-truth items is well-structured and aligns with the recall optimization objective.

---

> ### Author Rebuttal · Authors · 2025-04-01
>
> Thanks very much for the detailed and insightful review.
>
> For the questions:
>
> 1) In cascade ranking systems, **we can often explicitly define the ground-truth, thus optimizing the end-to-end recall directly maximizes selection efficiency. So we treat it as the golden metric**. Other intermediate metrics, such as precision or diversity, are less critical in this context. Recall and NDCG for single stage are also intermediate metrics for observation and analysis.
> 2) All baseline results were obtained by re-implementing or adapting the source code under the same experimental conditions as our proposed LCRON, rather than directly citing results from previous papers. **Since none of the baseline methods have been evaluated on the RecFlow dataset under cascade ranking settings. For FS-RankNet & FS-LambdaLoss, we adapt standard implementations from the TF-Ranking library to PyTorch versions. For other baselines, when open-source code was available and runnable, we used it directly; otherwise, we implemented the baselines based on the descriptions in their respective papers**. All methods were evaluated using the same common hyperparameters (lr and batchsize, optimizer, initialization method, etc.) to ensure fair comparison.
> 3) For the public experiments, the training data organization for the two-stage cascade ranking is described in lines 284-297. There is no additional data pre-processing step. I assume you might be asking how the data filtering is performed in a cascade ranking system, i.e., how many items the retrieval model selects to pass to the ranking model and how many items the ranking model then selects as the final output. These specific settings are detailed in lines 298–312 of the draft.
> 4) **Yes, we performed a grid search on the main hyperparameters for all methods to ensure fair comparisons. For the baselines, we reported the best results, and for LCRON, we included a sensitivity analysis in the appendix. Specifically, the parameters we tuned include: temperature for ICC (0.05,0.1,0.5,1.0), tau for ARF and LCRON (1,20,50,100,200,1000); alpha (0,0.25,0.5,0.75,1) for RankFlow; and top-k (10,20,30,40) and smooth factor (0,0.25,0.5,0.75,1) for FS-LambdaLoss**. BCE and FS-RankNet do not have independent hyperparameters. Regarding the learning rate (lr) and batch size, since all methods used the same setting for fair comparison and results on industrial applications are typically not very sensitive to these hyperparameters, we did not experiment with different lr and batch sizes. **Considering your concern, we ran an additional four sets of experiments to validate performance under different LR and batch sizes, rerunning all eight methods for each set, resulting in 32 experimental runs**. Due to time limitations, each method was run only once. Statistical significance can be assessed by referring to the mean±std and p-value from other significance tests. Due to space limitations, we only show the End-to-End Recall under different LR and batch sizes:
>
> |Method|bs=512,lr=0.001|bs=2048,lr=0.001|bs=512,lr=0.02|bs=2048,lr=0.02|
> |-----|-----|-----|------|------|
> |BCE|0.8181|0.8106|0.8637|0.8582|
> |ICC|0.7644|0.7459|0.4972|0.5061|
> |RankFlow|0.8326|0.8166|0.8768|0.8722|
> |FS-RankNet|0.7537|0.7533|0.7935|0.7884|
> |FS-LambdaLoss|0.8289|0.8194|0.8777|0.8726|
> |ARF|0.8288|0.8174|0.8704|0.8667|
> |ARF-v2|0.8302|0.8202|0.8776|0.8725|
> |LCRON|**0.8396**|**0.8247**|**0.8841**|**0.8785**|
>
> **It can be seen that our method achieves consistently optimal results, demonstrating the robustness of our approach**. These sensitivity analysis details and the hyperparameter tuning specifics for the baselines will be added to the appendix.
>
> 5) Please refer to our response to Reviewer VkMd (for weakness 1).
>
> 6) Please refer to our response to Reviewer VSAL. The improvement of LCRON over other baselines is statistically significant.
>
> For the missing reference:
>
> Thank you for highlighting these works. [1] is already introduced in related work. [1] and [3] are differentiable sorting methods, but they don't produce the permutation matrix, making them incompatible as foundation components for LCRON compared to NeuralSort and SoftSort. We will discuss this in related work. Our current comparisons include state-of-the-art single-stage recall optimization methods (e.g., ARF) and joint learning approaches for cascade ranking (ICC, RankFlow, FS-RankNet, FS-LambdaLoss). [2] and [4] focus on optimizing ranking metrics (e.g., NDCG and Precision) for single-stage models, which differ from our end-to-end cascade ranking objective. We will explicitly highlight this distinction in the related work section.
>
> Due to space limitations, we only show the key results (e.g., end-to-end recall) in the rebuttal text. Full additional results can be found in this anonymized github link: https://anonymous.4open.science/r/2025038594/
>
> If you have any further questions or concerns, we will make every effort to provide further clarification.

---

### Decision · Program_Chairs · 2025-05-01

**Decision:**

Accept (poster)

**Comment:**

This paper presents LCRON, a principled end-to-end training framework for cascade ranking that introduces a novel surrogate loss aligning stage-wise objectives with the global top-k selection goal. The well-motivated submission addresses key limitations in current multi-stage ranking systems, namely inter-stage misalignment and training-test discrepancies. The theoretical formulation is sound, and the experimental results are strong, demonstrating consistent improvements over state-of-the-art baselines across both public and industrial benchmarks. The reviewers highlight the paper’s practical relevance, methodological novelty, and strong empirical validation. Final scores reflect a consensus leaning toward acceptance, with three reviewers recommending weak acceptance and two recommending clear acceptance. I recommend acceptance of this paper if there is room.